# Genomic and Epigenomic Changes in the Progeny of Cold-Stressed *Arabidopsis thaliana* Plants

**DOI:** 10.3390/ijms25052795

**Published:** 2024-02-28

**Authors:** Ashif Rahman, Narendra Singh Yadav, Boseon Byeon, Yaroslav Ilnytskyy, Igor Kovalchuk

**Affiliations:** 1Department of Biological Sciences, University of Lethbridge, Lethbridge, AB T1K 3M4, Canada; ac.ashif.rahman@gmail.com (A.R.); nsyadava2004@gmail.com (N.S.Y.); slava.ilyntskyy@uleth.ca (Y.I.); 2Biomedical and Health Informatics, Computer Science Department, State University of New York, 2 S Clinton St, Syracuse, NY 13202, USA; boseon.byeon@oswego.edu

**Keywords:** multigenerational cold stress, stress memory, *Arabidopsis thaliana*, intergenerational inheritance, epigenetic variations, genetic variations, SNPs, InDels

## Abstract

Plants are continuously exposed to various environmental stresses. Because they can not escape stress, they have to develop mechanisms of remembering stress exposures somatically and passing it to the progeny. We studied the *Arabidopsis thaliana* ecotype Columbia plants exposed to cold stress for 25 continuous generations. Our study revealed that multigenerational exposure to cold stress resulted in the changes in the genome and epigenome (DNA methylation) across generations. Main changes in the progeny were due to the high frequency of genetic mutations rather than epigenetic changes; the difference was primarily in single nucleotide substitutions and deletions. The progeny of cold-stressed plants exhibited the higher rate of missense non-synonymous mutations as compared to the progeny of control plants. At the same time, epigenetic changes were more common in the CHG (C = cytosine, H = cytosine, adenine or thymine, G = guanine) and CHH contexts and favored hypomethylation. There was an increase in the frequency of C to T (thymine) transitions at the CHH positions in the progeny of cold stressed plants; because this type of mutations is often due to the deamination of the methylated cytosines, it can be hypothesized that environment-induced changes in methylation contribute to mutagenesis and may be to microevolution processes and that RNA-dependent DNA methylation plays a crucial role. Our work supports the existence of heritable stress response in plants and demonstrates that genetic changes prevail.

## 1. Introduction

Environmental factors actively influence the growth as well as the reproductive and biological fate of plants. Plants must endure environmental stresses during most of the time in their life cycle because they cannot escape them. Since plants are frequently exposed to various environmental stresses, the molecular- and cellular-level changes in the physiology and morphology observed in plants grown under stressed conditions mostly disappear when stress conditions are no longer present [1]. However, some residual level of response is often present, allowing plants to respond to stress more efficiently, and this is known as somatic stress memory, priming or acclimation [2,3].

Most of the environmentally induced memories are relatively short and can exist only as somatic memories. Occasionally, somatic stress memory persists to the next generation, resulting in changes in genome stability, plant morphology and stress tolerance [4,5,6,7]. This process is known as intergenerational stress response. And even less frequently, the memory of stress exposure is carried yet to another generation, without stress exposure, resulting in so called transgenerational stress response [7,8]. It has been well documented that a transgenerational memory can play a role in generating epigenetic variants, in the form of differential DNA methylation or/and histone modifications that can allow plants to exhibit a certain degree of tolerance to the environmental stresses and consequently lead to adaptation [2,9].

Abiotic stresses can trigger changes in methylation patterns, genomic stability, and stress tolerance in the progeny. For instance, in the progeny of salt-stressed *Arabidopsis thaliana* plants, changes have been reported in the genome stability, DNA methylation, histone modifications, and gene expression [10,11]. Arabidopsis plants exposed to cold stress exhibit changes in the transposon expression and recombination frequency [12]. In response to heat stress, changes in the phenotypes, genotypes and epigenotypes have already been reported in *Arabidopsis thaliana* [7,13]. Several studies in *Arabidopsis thaliana* suggest that environmental stresses may also lead to an increase in the genomic diversity in the plants’ progeny, even in the untreated generations, and they potentially result in the adaptation to adverse conditions [7,14,15,16,17]. 

Inter- and transgenerational effects involve passing the information about responses to changes in the environment, including physical, chemical and biological (pathogens) exposures from parents to offspring [18]. Mechanisms of intergenerational inheritance may be very versatile and involve primary and secondary metabolites, including differentially expressed mRNA and non-coding RNAs, proteins, peptides and other small and large organic molecules [18]. In contrast, transgenerational effects likely only involve differential DNA methylation, histone modifications and changes in chromatin structure, but could also involve non-coding RNAs, providing there are mechanisms of their amplification [19]. Several studies suggest that the reprogramming of phenotypes of the offspring can be inherited through epigenetic mechanisms [20,21]. The environmentally induced and inherited epigenetic marks can facilitate plants’ adaptation to the changing environments. It can for example lead to microevolution in clonal plants in a relatively short time [22]. 

Epigenetic mechanisms in plants primarily consist of DNA modifications, e.g., DNA methylation, small non-coding RNAs and histone modifications. DNA methylation is environmentally inducible and, in many cases, inheritable [2]. However, due to the reprogramming of the environmentally induced epigenetic marks in meiosis, in most of the cases, epigenetic modifications are maintained within generations and infrequently passed onto the sexually derived offspring [23]. DNA methylation occurs in CG, CHG and CHH contexts in plants where C is a cytosine, G is a guanine and H represents the nucleotides A, T or C. CHH methylation is relatively unstable since it is asymmetrical and can primarily be maintained via the guidance of non-coding RNAs, such as small interfering RNAs (siRNAs) through the RNA-dependent DNA methylation (RdDM) mechanism [19]. In plants, epigenetic regulation such as DNA methylation is meiotically stable and can be transmitted either through maintenance methyl transferases at symmetrical cytosines (CG and CHG) or at asymmetrical CHH via small RNAs, as stated above. For the successful transmission of the transgenerational information to progeny, epigenetic marks need to be transmitted by the settings of genome reprogramming during gametogenesis and zygote development [24]. 

Fluctuations in the environmental conditions, associated with climate changes put enormous pressure on our ability to grow adequate amount of crop. It is therefore important to understand how plants establish the trans- and intergenerational responses and what occurs in the progeny of stress-exposed plants. In this work, we have cold-stressed *Arabidopsis thaliana* (ecotype Columbia) plants for 25 generations. We analyzed phenotypic, genetic and epigenetic changes and response to stress in the first and 25th generations of plants grown in normal conditions and cold-stressed. We hypothesized that the progeny of plants exposed to cold stress across 25 generations would be genetically and epigenetically more diverse than the parental plants. Indeed, we found that they are. We found these changes to be non-random and associated with the stress-related genes and pathways. 

## 2. Results

### 2.1. Comparison of Genomic Variants

In this work, we produced 25 generations of cold exposed plants, starting from the seeds from one single homozygous *A. thaliana* (Columbia ecotype) plant. We propagated plants for 25 generations either at normal temperature (22 °C) or by exposing them to cold (4 °C)—see Methods for details. In this experiment, we used the seeds from control generation 2 (F2C), control generation 25 (F25C) and cold-exposed generation 25 (F25Cd). 

To analyze the rate of occurrence of genomic variants, we conducted whole genome sequencing on five individual plants from each group, F2C, F25C and F25Cd. There was on average 10 mln reads in each individual sample and over 97% of reads mapped to *A. thaliana* genome; no significant difference in the read number of mapped reads were observed between F2C, F25C and F25Cd groups.

The average number of single nucleotide polymorphisms (SNPs) calculated from the five biological replicates was significantly (*p* < 0.05) higher in the cold stressed progeny (F25Cd) than in the parallel (F25C) and parental (F2C) control progeny (Figure 1A). F2C and F25C were not different from each other. 

The average number of insertions and deletions (InDels) calculated from five biological replicates was also significantly higher (*p* < 0.05) in the cold stressed progeny (F25Cd) compared with the parallel and parental control progeny (Figure 1B). The F2C and F25C groups were similar. The difference was primarily attributed to deletions, as number of insertions was statistically similar (*p* > 0.05) (Figure 1C,D). 

#### 2.1.1. Unique and Common SNPs and InDels 

We then analyzed the number of common and unique SNPs and InDels in F2C, F25C, and F25Cd. We found that F25Cd group had nearly double the number of unique SNPs as compared to other groups (Figure 2A). The same was found for the InDels (Figure 2A).

#### 2.1.2. Analysis of Distribution of SNPs and Their Potential Impact

Mutations are rarely evenly distributed in the genome, with some regions/chromosomes having higher rate. Our analysis of distribution of SNPs showed that variant rates were the highest in mitochondrial genome, while lowest in chromosomes 1, 2 and 5 (Figure 2B). There was a change in the variant rates in response to cold stress—a decrease in mitochondrial genome and an increase in chromosomes 3, 4 and 5 Specifically, in mitochondrial genome the rate in F25Cd decreased significantly from one variant per 6186 nt in F2 and one variant per 6336 nt in F25C to one per 6600 (*p* < 0.05). In contrast, the variant rates increased significantly in chromosomes 3, 4 and 5. 

We then analyzed the SNPs by their potential impact in the genome, such as amino acid change, change to start/stop codon, change in the sequence of regulatory element etc. According to SnpEff, all SNPs are classified into modifiers, low, moderate, and high impact. “High impact” would be the most deleterious, while “modifiers” would be the least, and would likely have only marginal effect on gene expression. It was found that F25Cd had a decrease in the percentage of low, moderate and high impact SNPs, while they had an increase in the percentage of “modifier” SNPs (Figure 2C–F). This is an interesting phenomenon; it appears that the increase in the number of SNPs in F25Cd plants comes primarily at the expense of mutations that are least impactful on gene expression. 

We next analyzed whether SNPs represent missense/nonsense and non-synonymous/synonymous mutations. We found that F25Cd group had significantly higher percentage of missense mutations as compared to F25C, but was similar to F2 group (Figure 3A). Percentage of nonsense mutations was similar across three groups (Figure 3B). Aso, F25Cd group had more non-synonymous and less synonymous mutations, as compared to F25C group, while, again, being similar to F2 group (Figure 3C,D).

We then analyzed the type of mutations by a nucleotide change. We found C to T and G to A mutations being most abundant in all groups (Figure 4). F25Cd group was significantly (*p* < 0.05) different from both F25C and F2 groups in the percentage of A/C, G/A, T/A, C/G and C/T mutations (Figure 4). F25C was significantly (*p* < 0.05) different from both groups in G/C and T/G mutations, while F2 was different from both groups in A/T mutations (Figure 4). F25Cd had higher frequency of G/A, C/G and C/T mutations and lower frequency of A/C and T/A mutations as compared to other two groups. C to T transitions appeared to be the most common type of mutations associated with deamination of methylated cytosines [26]. The fact that the frequency of these mutations increases in F25Cd suggests that environment-induced changes in methylation contribute to mutagenesis. When we analyzed mutations of cytosines at specific contexts, we found that C to T mutations at the CpG islands were predominant of three possible mutations (C to T, C to A and C to G), being over 2/3 of all mutations at these sites (Appendix A). There was no significant difference between F2, F25C and F25Cd in any of the three types of mutations at the CpGs (Appendix A), indicating that likely the increase in C to T mutations in F25Cd comes from mutations occurring at CHH sites. 

To analyze whether SNPs genomic position is random or not, we analyzed the location of SNPs in different genomic regions. We noticed a decrease in the SNP percentages in the introns and splice-donor/splice-acceptor sequences in F25Cd group, but the differences between this group and F25C and F2 were not significant (Appendix A).

We next analyzed the rate of transition (Ti) and transversion (Tv) mutations and the ratio of transitions to transversions (Ti/Tv). Transition mutations, change from purine to purine or pyrimidine to pyrimidine, are very common. Mathematically, for every Ti mutation, there are 2 Tv mutations, giving Ti/Tv ratio of 0.5; but in most organisms, this ratio is heavily skewed towards Ti mutations, with Ti/Tv of close to 2.0 being very common [27,28]. Our analysis showed the significant increase in Ti/Tv ratio in F25Cd group, stemming from the increase in Ti and decrease in Tv mutations (Figure 5). Analysis of Ti, Tv and Ti/Tv ratios in different genomic locations showed an interesting result. Ti/Tv ratio was significantly (*p* < 0.05) higher in the 150 nt downstream region in F25Cd as compared to other groups (Figure 5D).

#### 2.1.3. SNPs and InDels Associated with Transposons

Transposable elements are frequently activated in response to stress. Previous reports show evidence of changes in the regulation and activity of transposons in the progeny of plants exposed to stress [29]. We found that F25Cd group had significantly (*p* < 0.05) higher number of SNPs associated with transposons, as compared to F2 and F25C (Figure 6A). F2 and F25C group were statistically similar (*p* > 0.05). We then checked what regions of transposons are associated with higher SNPs presence in F25Cd and found that it was higher in the promoter and gene body, but not at the enhancer regions of transposons (Figure 6B–D). 

Same analysis performed for InDels showed that there was a significant increase in F25Cd as compared to both groups (Figure 6E). Analysis of various parts of the transposons showed the higher numbers for F25Cd group, but none of them were significantly different from F2 and F25C (Figure 6F–H).

#### 2.1.4. The Functional Classification of SNPs and InDels

To analyze the potential impact of differences in SNPs, the functional classification of genes associated with these SNPs was performed. Enrichment in three categories was analyzed: biological process, molecular function, and cellular component. Enrichment would represent a statistically significantly higher or lower number of SNPs in the specific category as compared to a random occurrence in the *A. thaliana* genome.

Comparison of F25Cd to other groups showed that biological process “protein metabolism” was uniquely under-represented, as compared to other groups (Table 1). Biological process “transcription, DNA-dependent” was overrepresented in F2 group alone. All other enriched biological processes were similar between F25Cd and other groups.

Molecular function “transcription factor activity” was enriched in F2 group but not in the progeny groups. F25Cd group was significantly different from the other two groups in molecular functions “other binding” and “transferase activity”.

Finally, comparison of cellular components showed that F25Cd was uniquely enriched in “other cellular components”, “mitochondria” and “extracellular” (Table 1).

The same analysis for InDels showed that among biological processes, “developmental stimuli” and “response to abiotic or biotic stimulus” were significantly different in F25Cd from other groups and “response to stress”, “cell organization and biogenesis” and “other biological processes” were uniquely enriched in F25Cd groups (Table 2). In molecular function category, “nucleotide binding” was significantly different and “kinase activity” uniquely enriched in F25Cd as compared to other groups. Finally, among cellular components, “cytosol” was significantly different and “plastid” and “cell wall” were uniquely enriched in F25Cd as compared to the other groups (Table 2).

### 2.2. Epigenomic Profiling

Whole Genome Bisulfite Sequencing (WGBS) was used to profile the epigenomic changes in the form of changes in DNA methylation patterns. For epigenomics, the distribution of Differentially Methylated Cytosines (DMCs) and Differentially Methylated Regions (DMRs) was analyzed to study responses to cold stress over multiple generations. 

#### 2.2.1. The Percentage of Global DNA Methylation 

Bisulfite sequencing data revealed that the average percentage of global genome methylation was significantly higher in F25Cd in the case of the CpG, while statistically similar in CHG and CHH contexts, as compared to the other two groups. The average percentage of global genome methylation in the CpG context was 23.46%, 23.64%, and 26.14%, respectively, in F2C, F25C, and F25Cd (Figure 7). Methylation at CHG in F25Cd group was higher than in the other two groups, but the difference was not significant. CHH global methylation was lower in the progeny of cold as compared to F2C, but the differences were not significant (Figure 7C).

#### 2.2.2. Analysis of DMCs and DMRs 

The analysis of the total number of DMCs showed that the largest number of DMCs was observed in F25C vs. F2C comparison group—80,464. Other comparisons showed 63,524 in F25Cd vs. F2C group and 77,648 in F25Cd vs. F25C group (Figure 8A). This was somewhat surprising, as we hypothesized that there would be more changes in methylation levels observed in the progeny of stressed plants as compared with the controls.

The analysis of DMCs in a specific sequence context revealed a different pattern. While DMCs in the CpG context were similar in all comparison groups, there were more DMCs in CHG and CHH context in F25Cd group compared to F25C or F2C groups (Figure 8B–D). In fact, over a 10-fold difference was observed in the hypomethylated DMCs in the CHH context in F25Cd vs. F2C as compared with F25C vs. F2C—1894 vs. 176 DMCs, respectively (Figure 9D). A similar picture was observed in the CHG context, where over a 5-fold difference was observed between the two comparisons—1189 vs. 230 DMCs (Figure 8C). These data suggest that the cold stress-induced epigenetic variations at single cytosines are primarily associated with changes in the CHG and CHH contexts.

Methylation frequently has more significant effect when occurs on multiple cytosines, such as found in CpG islands. We thus decided to check the methylation occurrence in 100 and 1000 bp windows. The analysis of the total number of DMRs in a 100-bp window did not show any substantial difference between the comparison groups; there were 12,619 DMRs in F25C vs. F2C comparison and 10,722 DMRs in F25Cd vs. F2C group as well as 14,524 DMRs in F25Cd vs. F25C group (Figure 8E). DMRs were similar in CpG context (Figure 8F). There were more DMRs in CHG context (Figure 8G). In CHH context, there were substantially more hypermethylated DMRs in the comparison groups involving F25Cd group (Figure 8H). This analysis showed that DMRs in 100-bp window are also different primarily at CHG and CHH contexts.

The analysis of the total number of DMRs in a 1000-bp window showed more DMRs in F25Cd comparison groups: 86 DMRs in F25Cd vs. F2C group, 67 DMRs in F25Cd vs. F25C group and 32 DMRs in F25C vs. F2C group (Figure 8I). There were more DMRs in both, CpG and CHG contexts in the F25Cd comparison groups (Figure 8J,K). There was not enough data for CHH context to plot the comparison.

#### 2.2.3. Hierarchical Clustering of Groups in DMCs and DMRs

Clustering of DMCs (Figure 9) and DMRs (Appendix A) in F25Cd vs. F2C comparison group showed a clear separation of F25Cd samples and F2C samples in all three methylation contexts. In contrast, in the F25Cd vs. F25C group comparison (Figure 9B,D), F25Cd samples were not as clearly separated from F25C group, especially in CG and CHG context. 

Similarly, the analysis of the relatedness of the samples using heatmaps and hierarchical clustering done for DMCs in the CG context demonstrated clear separation between F25Cd and F2 groups for all five samples (Figure 10). Similar comparison between F25Cd and F25C also showed a clear separation for 4 out of 5 analyzed samples. Finally, comparison between F25C and F2 groups showed separation for 3 samples out 5 in hypermethylated group and 2 out of 5 in hypomethylated, with the other samples being more similar (Figure 10). Analysis of heat maps in DMRs (100 bp window) in the CG context showed the same picture as DMCs analysis (Appendix A).

Analysis performed for CHG and CHH contexts showed similar trend, although, the separation was less clear than in the case of CG contexts, especially in the hypermethylated group (Appendix A). 

#### 2.2.4. The Analysis of the Distribution of DMCs across the Chromosomes 

The analysis of the distribution of DMCs across five *A. thaliana* chromosomes confirmed that the differences between F25Cd and F2 groups are primarily observed in hypomethylated cytosines in CHG and CHH contexts (Appendix A). 

#### 2.2.5. Distribution of DMCs and DMRs in the Genomic Regions

To decipher the impact of all observed differential methylation events, DMCs were further characterized to determine whether they were preferably located near genes. The location of hypo- or hypermethylated DMCs was compared to the annotated Arabidopsis genes using genomation. 

The percentages of hypermethylated DMCs in CG context (F25Cd vs. F2C, F25Cd vs. F25C and F25C vs. F2C) were the highest in the exon region (37%, 44% and 47%, respectively), then in the promoter region (34%, 26% and 30%, respectively), followed by the intron region (11%, 15% and 12%, respectively) and the intergenic region (19%, 13% and 11%, respectively) in CpG sites (Figure 11, upper panel). 

We noted underrepresentation of hypermethylation in exon region and overrepresentation in promoter and intergenic regions in F25Cd vs. F2C, as compared to other groups. No such differences were observed for the hypomethylated DMCs.

In the case of CHG sites (Figure 11, middle panel), all groups (F25Cd vs. F2C, F25Cd vs. F25C and F25C vs. F2C) showed the highest percentage of hypermethylated DMCs in the promoter (49%, 47% and 53%, respectively), then in intergenic regions (41%, 46% and 36%, respectively), followed by exon (8%, 5% and 7%, respectively) and intron (2%, 2% and 3%, respectively) regions. Somewhat lower percentage of methylation in the promoter was in the F25Cd vs. F2C and F25Cd vs. F25C groups; this was compensated by higher percentage of hypermethylation in the intergenic region. Interestingly, F25Cd vs. F2C and F25Cd vs. F25C groups showed the highest percentage of hypomethylated DMCs in intergenic regions (61% and 60%, respectively), followed by the promoter (35% and 34%, respectively) as compared with F25C vs. F2C group, where the hypomethylated DMCs in the intergenic and promoter regions occurred at 44% and 52%, respectively (Figure 11, middle panel). Thus, it appears that in CHG contexts, there was a major shift towards the decreased hypomethylation at the promoter region and increased hypomethylation in the intergenic region in the F25Cd group as compared to either control group.

In the CHH context, the percentage of both hyper- and hypomethylated DMCs was higher in the promoter region, followed by the intergenic region, exon and then intron regions in all comparison groups (Figure 11, lower panel). We noted a substantial decrease in the percentage of hyper- and hypomethylated cytosines in the promoter of F25Cd group; F25Cd vs. F2C and F25Cd vs. F25C had almost 20% lower percentage of methylation at the promoter as compared to F25C vs. F2C group. This came at the expense of the increase in the percentage of methylation at the intergenic regions. F25Cd group also had lower percentage of hypermethylated DMCs at the exons.

In DMRs, somewhat similar picture was observed. At the CG context, increase in the percentage of hypermethylated cytosines was observed in the promoter and intergenic regions, and a decrease in the exon regions in the F25Cd plants (Appendix A). At the CHG and CHH contexts, there was a decrease in the percentage of hypomethylated cytosines in the promoter and an increase in the intergenic region in F25Cd plants as compared to controls. 

#### 2.2.6. Biological Enrichment Analysis of DMCs and DMRs 

The functional classification of variants, DMRs and DMCs unique to each test group, was interpreted using SuperViewer to identify regions with the statistically over-represented numbers of genes and genomic features.

The biological enrichment analysis revealed that CG hyper- and hypomethylation in DMC (Figure 12) and DMR (Appendix A) sites were enriched in many general and specific biological processes, but none of them was significantly different in comparison of F25Cd group to any other group.

In the hypermethylated DMCs in the CHG context, F25Cd group was significantly enriched in “response to abiotic and biotic stimuli”, “response to stress”, “other biological processes”, “other cellular processes”, “other metabolic processes”, “protein metabolism”, “transcription, DNA-dependent” and “developmental processes”, while comparison of F25C group to F2C group only showed enrichment in “signal transduction” (Figure 12). Similar biological processes were enriched in the hypomethylated DMCs in the CHG context.

In the hypermethylated DMCs in the CHH context, F25Cd group was significantly enriched in “other cellular processes”, “transport” and “other metabolic processes”, while in the hypomethylated, F25Cd group was enriched in “response to abiotic and biotic stimuli”, “response to stress”, “electron transport of energy”, “other biological processes”, “other cellular processes”, “other metabolic processes” and “developmental processes”, while F25C comparison to F2C only had enrichment in “protein metabolism” and “transcription, DNA-dependent”. 

## 3. Discussion

Being sedentary, plants are more vulnerable to environmental changes. Plants face several external stresses such as abiotic stresses, including chemical and physical changes, changes in light intensity, temperature fluctuations, nutrients and water availability, wind and other mechanical stimuli, and biotic stresses, including various pathogens [30,31]. Environmental stresses have an impact on the directly exposed generations as well as on their progeny [32]. This study aimed to understand the multigenerational cold-stress effects on plant genotypes and epigenotypes. 

### 3.1. Genomic Analysis 

Environmental stresses can be mutagenic and capable of causing genome instability [33,34]. Environmental changes may also increase homologous recombination events or can facilitate the mobilization of transposable elements [7,35]. Eventually, stress may induce changes in the genetic material and increase the chances of genomic diversity leading to adaptation [36]. 

#### 3.1.1. Single Nucleotide Polymorphisms 

SNPs are vital genetic variations that can directly disrupt gene function and affect plant adaptability in the changing environment [37]. For instance, SNPs can affect light response and flowering time by changing amino acids in phytochromes A and B [38]. In our study, genomic analysis revealed that the number of SNPs variants were significantly higher in the cold-stressed progeny. These SNPs were not evenly distributed in the genome; most changes were observed in chromosomes 3, 4 and 5 and mitochondrial DNA—higher frequency of SNPs was found in chromosomes 3, 4 and 5, and lower in the mitochondrial genome. Our recent reports on the 25 generations of plants exposed to heat also indicated higher number of SNPs in the progeny of stressed plants [7]. Zhang et al. (2013) found a frequent occurrence of SNPs in drought-resistance genes in common wheat. They also found that SNPs were associated with the genes responsible for the developmental processes and abiotic stress resistance in wheat [39]. SNPs can also create new splice sites and alter gene function [40]. Therefore, a higher number of SNPs in the cold-stressed progeny may be an indication of the adaptive processes occurring in the progeny of stressed plants.

It is hard for us to comment whether the observed rate of mutations is especially high in our experimental set-up. Comparison of mutation rate with other studies would be very difficult, as in our case, it was a cumulative rate over 25 generations. We did not calculate the rate per generation, so we can not compare it with other reports—most report mutation rate per nt per generation. We also do not think that the *Arabidopsis thaliana* cultivar used in this study had unusually high level of mutations, as it was used in many publications previously [7,14,15,16,17].

Analysis of the potential impact of SNPs on the gene expression showed that F25Cd were enriched in mutations that are least impactful on gene expression, suggesting selection against more impactful mutations. It is possible that there exists some sort of purifying selection in the progeny of cold-stressed plants. Recent report supports our finding, suggesting that mutation occurrence in *A. thaliana* genome is directional—mutations are less frequent in gene bodies and in essential genes; in addition, the authors showed that ~90% of all mutations in *A. thaliana* genome were driven and defined by epigenetic modifications, and thus were non-random [41].

Analysis of types of SNPs that occur showed that F25Cd had higher frequency of several types of mutations, including G/A, C/G and C/T mutations. The latter is one of the most common mutations associated with deamination of methylated cytosines [28]. The increased rate of C to T mutations in F25Cd could suggest that there was an increase in the number of methylated cytosines in plants grown in the presence of cold and that these methylated cytosines were mutated. Further analysis showed that C/T mutations at the CpG islands were similar across three groups, suggesting the increase in C to T mutations in F25Cd likely comes from mutations occurring at CHG and CHH sites. Considering that methylation at asymmetrical cytosines is maintained by the activity of RdDM [19], it can be suggested that transgenerational changes in the progeny cold-stressed plants are maintained by non-coding RNAs acting at the non-symmetrical cytosines.

Ti/Tv ratio appear to be relatively stable when organisms are grown in stable environment, but have a tendency to change in response to stress [42]. Our work demonstrated a significant increase in Ti/Tv ratio in F25Cd group, due to the increase in Ti and decrease in Tv mutations. Specifically, 150 nt downstream region in F25Cd had significantly higher ratio of Ti/Tv ratio as compared to other groups (Figure 5D). Such bias may reflect the importance of specific nucleotide composition in this downstream region of the gene for the regulation of gene expression.

#### 3.1.2. Insertions and Deletions

In our study, we have found that the number of InDels is higher in the cold-stressed progeny (F25Cd) as compared with the control progenies, F2C or F25C. The increase was attributed to the number of deletions rather than insertions. Similar phenomenon was found in the progeny of plants exposed to heat for 10 generations [43]. InDel variants could impose potential effects on the genome, affecting chromatin structure, gene expression and response to stress [44]. InDels can be by far more deleterious than mutations, and thus would be eliminated at higher rate than SNPs. It the same time, those with neutral or positive effect would be fixed much faster. Therefore, higher numbers of InDels in the stressed progeny can be an indicator of genomic diversity since InDels can be considered as a crucial factor shaping the evolution of the genomes and species. 

We have also noticed that the increase in the average number of transposons associated with SNPs and InDels in the cold-stressed progeny (F25Cd) as compared with the control progenies. Our previous studies showed activation of transposons in the immediate progeny of plants exposed to cold [12], heat [13] and UVC [29]. It has been reported that transposable elements can act as stress-responsive regulators by controlling gene expression [45]. Transposons are believed to have played a crucial role in evolution of species. Current work demonstrates that not only transposons are activated by stress in the progeny, but they also appear to undergo rapid evolution (increased SNPs and InDels). 

#### 3.1.3. The Functional Classification of SNPs and InDels

An enrichment analysis of SNPs revealed the over-representation of “mitochondria” and “extracellular” cellular components in the cold-stressed progeny. This suggests that mutations more frequently occur in these regions in the cell. This correlates with our finding that F25Cd is different from other groups in mutations in mitochondrial genome. Extracellular matrix is involved in cell-cell and cell-environment interaction, and thus is the first frontier that encounters many stresses. Increased rate of SNPs in genes encoding components of extracellular matrix may be adaptive.

Analysis of the InDels showed that biological processes “developmental stimuli”, “response to abiotic or biotic stimulus”, “response to stress” and “cell organization and biogenesis” were significantly different in F25Cd from other groups. In molecular function category, “nucleotide binding” was significantly different and “kinase activity” uniquely enriched in F25Cd as compared to other groups. Changes in kinase activity is an important response to stress. Finally, among cellular components, “plastid” and “cell wall” were significantly enriched in F25Cd as compared to the other groups (Table 2). Again, changes in the mutations in genes involved in plastid (likely chloroplasts) metabolism could also be beneficial for stress adaptation. Similar to our study, Wang et al. (2017) reported a total of 211 differentially expressed proteins due to cold stress response, where the over-representation was observed in protein metabolism and translation, stress responses, the membrane, and transport processes [46].

### 3.2. Analysis of Methylome

Finally, we have carried out whole-genome bisulfite sequencing (WGBS) of the control (F2C and F25C) and cold-stressed (F25Cd) plants to decipher epigenomic variations among the tested generations. DNA methylation is the most studied and best-narrated epigenetic modification for decoding the mechanisms of gene expression and the status of the chromatin structure [47]. Immediate response to stress involves a multitude of epigenetic changes, from DNA methylation and histone modifications to nucleosome repositioning and expression of ncRNAs [48]. Several studies suggested that multigenerational exposure compared with the exposure of a single generation could cause substantially higher number of heritable epigenetic variations [49,50,51].

#### 3.2.1. The Percentage of Global DNA Methylation 

We analyzed the epigenome of plants in the control and cold-stressed progeny in the light of previous studies that reported hyper- and hypomethylation in plants under stress [52,53]. The methylome analysis revealed that the average percentage of global genome methylation in the CpG and CHG contexts was higher in the cold-stressed progeny, F25Cd. But in the CHH context, F25Cd showed the lower global methylation. Similar to the results of our study, Jiang et al. (2014) showed that soil salinity stressed lineages accumulate more methylation at the CpG sites than the control progenies [54]. Multigenerational exposure to heat resulted in lower global genome methylation at both CHG and CHH, while no effect was observed on CpGs [7]. 

#### 3.2.2. The Total Number of DMCs and DMRs 

We observed that the total number of DMCs and DMRs was higher in the control group F25C vs. F2C as compared to the stressed progeny groups F25Cd vs. F2C and F25Cd vs. F25C. This was somewhat surprising as we hypothesized that there would be more epigenetic changes in the progeny of stressed plants (F25Cd) as compared with the parallel and parental control progenies. Similar phenomenon was observed in the progeny of plants exposed to heat [7]. As we observed the higher rate of mutations, and specifically C to T transitions in F25Cd group, we can hypothesize that a number of methylated cytosines in this group were mutated, thus decreasing an overall methylation rate in this group. But since mutations are still rare, it is unlikely this would account for the observed difference to the other groups. It is also plausible to think that fewer DMCs/DMRs in the progeny can be an adaptation trait in response to cold stress where stressor might reduce the rate of spontaneous/stochastic epimutations.

Analysis of DMCs and DMRs in the specific sequence contexts showed that they were similar at CG cytosines between F25C and F25Cd, but there was significantly more of them in CHG and CHH contexts in F25Cd group as compared to control groups. The data further confirm the important role of hypomethylation at CHG and CHH cytosines in cold stress adaptation. Similar results were achieved by Yadav et al. (2022) in *A. thaliana* progeny propagated at high temperature [7]. Exposure to salt for 11 generations also resulted in changes in DNA methylation that mostly occurred in CHH context; the authors found CHH to be hypermethylated at transposons, but also found that CG and CHG sites were significantly hypomethylated [51]. Similarly, the progeny of rice plants exposed to heavy metals showed hypomethylation at CHG sites [55]. In support of this concept, several reports in *A. thaliana* actually suggested that it is common for plants to methylate cytosines in the sequence contexts of CHG and CHH, which are mostly guided by small RNAs or the heterochromatin-directed methylation pathways [56,57]. These findings demonstrate that hypomethylation at CHG and CHH regions could be one of the adaptive responses to stress. 

#### 3.2.3. The Distribution of DMCs across Genic and Intergenic Regions

So, it appears that in F25Cd group, cytosines at the promoter region were hypermethylated at CG but hypomethylated at CHH context. In the exons, there was less hypermethylation at CG and CHH contexts in F25Cd group. In the intergenic regions, there was more hypermethylation at CG and CHH but more hypomethylation at CHG contexts. No changes were observed in the introns. Thus, changes in methylation in F25Cd group primarily occurred in the promoter and exon regions of the gene and in the intergenic area. Hypermethylation at the CpGs of the promoter may indicate tightening regulation of the gene expression, especially in tissue-specifically expressed genes. Hypomethylation at CHG and CHH regions especially in the exon area and intergenic regions associated with transposons may indicate the role of RdDM in the process and demonstrate adaptive nature of the process. Similar results were reported in the progeny of plants exposed to heat for 25 generations; in the CHH contexts, promoters were found to be hypermethylated, while intergenic regions and exons were hypomethylated as compared to control groups [7].

#### 3.2.4. Biological Enrichment Analysis 

In our study, the Gene Ontology analysis of differentially methylated regions revealed that all groups were relatively similarly enriched in the CG contexts, while significant differences were mainly observed in the CHG and CHH contexts. We found significant enrichment of “response to abiotic and biotic stimuli”, “response to stress”, “other biological processes”, “other cellular processes”, “other metabolic processes”, “protein metabolism”, “transcription, DNA-dependent” and “developmental processes” in the cold-stressed progeny compared with the control progenies. This set of comparisons demonstrated that methylation changes in F25Cd have the most significant effect in CHG and CHH context and target various biological processes, including those related to stress response. These biological functions and pathways are related to a normal plant development and associated with the stress response [58].

## 4. Materials and Methods

### 4.1. Plant Material

Seeds of *Arabidopsis thaliana* plants used for the experiment were obtained from a single inbred plant (Columbia ecotype, transgenic line 15d8). Two groups of plants, each consisting of 24 plants, were used. One group of plants was exposed to cold (12 h at 22 °C followed by 12 h at 4 °C for seven days, between days 7 and 14 after germination) for 25 consecutive generations (single generation is approximately three months, but it took us about 9 years to produce all 25 generations), while another group was propagated at normal conditions (16 h day (120 μmol photons m^−2^ s^−1^) at 22 °C and 8 h night at 20 °C); seeds from each generation were collected from 24 plants, pooled together and then ~100 seeds were used to start the next generation, where, after germination, 24 plants were randomly chosen to start another generation. Two independent stress groups and two independent control groups were generated and propagated independently for 25 generations, F1 to F25. The progeny of control plants were labeled as F1C through F25C, while the progeny of cold-stressed plants were labelled F1Cd through F25Cd.

### 4.2. Genomic and Epigenomic Profiling

The parental F2 plants and the progeny of F25 plants (both from the progeny of cold stressed plants and control plants) were grown to about 21 dpg, and then the leaf tissue was harvested for molecular analysis. We could not use F1 plants since we have run out of seeds. For methylation analysis, the generation F2C was considered as parental control progeny, F25C as parallel control progeny and F25Cd as the progeny of cold-stressed plants. The comparisons were done between F25Cd and F2C, F25Cd and F25C, as well as F25C and F2C. 

### 4.3. DNA Isolation

The whole rosette leaves of F2, F25C and F25Cd plants were collected, frozen in liquid nitrogen, and stored at −80 °C for DNA extraction. Total genomic DNA was extracted from approximately 100 mg of leaf tissue homogenized in liquid nitrogen using a CTAB protocol with some modifications. A DNA extraction buffer consisted of 31.8 g Sorbitol, 6 g Trizma base (Tris), 0.84 g EDTA in 500 mL of DDW, pH adjusted to 7.5 with HCl. A nucleic lysis buffer was prepared using 30.29 g Tris, 23.27 g EDTA, 73.05 g NaCl, 5 g CTAB dissolved in ~250 mL DDW, pH adjusted to 7.5. The total extraction buffer used was prepared with Na-bisulfite (38 mg/10 mL) added before use, 10 mL of nucleic lysis buffer, and 4 mL of 5% Sarkosyl. 700 µL of the total extraction buffer was used per sample. Samples were incubated at 65 °C for 1 h and inverted periodically. 700 µL of chloroform was added to the samples and shaken by hand for 5 min. The samples were centrifuged at 16,000× *g* for 10 min at 4 °C with a supernatant phase transferred to a new tube; this chloroform step was repeated. Two-third volume of isopropanol was added and incubated at room temperature for about 24 h to precipitate DNA. All samples were then centrifuged at 12,000× *g* for 15 min at 4 °C, the pellet of precipitated gDNA was rinsed twice with 70% ethanol and once in 100% ethanol, then air-dried at room temperature for about 10 min. 100 µL of P1 buffer (Qiagen Kit) mixed with RNAase was added, and the samples were incubated for 10 min at 37 °C.

An additional extraction was performed at this stage by adding 100 µL of phenol: chloroform mixture, centrifugation at 16,000× *g* for 10 min at 4 °C and transferring 90 µL of the supernatant phase to a new tube. 9 µL of Sodium acetate and 250 µL of 100% ethanol were added and incubated at room temperature to precipitate DNA. The samples were centrifuged at 12,000× *g* for 10 min at 4 °C to obtain pellets of precipitated DNA and washed twice with 1ml of 70% ethanol and once in 1 mL of 100% ethanol, then they were air-dried at room temperature. The DNA pellets were dissolved in distilled water and quantified using a NanoDrop 2000C spectrophotometer (Thermo Fisher Scientific Inc. Waltham, MA, USA). Also, agarose gel electrophoresis was performed to verify the integrity of DNA samples.

### 4.4. Whole Genome Sequencing (WGS) and Whole Genome Bisulfite Sequencing (WGBS)

The isolated genomic DNA was used for whole-genome sequencing (WGS) and whole-genome bisulfite sequencing (WGBS) (Illumina, San Diego, CA, USA) to assist in identifying the genomic and epigenomic (associated with changes in DNA methylation) profiles and variations. The WGS and WGBS libraries construction and sequencing have been carried out at the Centre d’expertise et de service Génome Québec, Montreal, Canada. For whole genome sequencing (WGS), gDNA was quantified using the Quant-iT™ PicoGreen^®^ dsDNA Assay Kit (Life Technologies, ThermoFisher Scientific, Waltham, MA, USA). 

#### 4.4.1. WGS Libraries Construction and Sequencing

For whole genome sequencing (WGS), gDNA was quantified using the Quant-iT™ PicoGreen^®^ dsDNA Assay Kit (Life Technologies). Libraries were generated using the NEBNext Ultra II DNA Library Prep Kit for Illumina (New England BioLabs, Whitby, ON, Canada) as per the manufacturer’s recommendations. Adapters and PCR primers were purchased from IDT. Size selection of libraries contained the desired insert size has been performed using SparQ beads (Qiagen, Toronto, ON, Canada). Libraries were quantified using the Kapa Illumina GA with Revised Primers-SYBR Fast Universal kit (Kapa Biosystems, Millipore Sigma Canada Ltd., Oakville, ON, Canada). Average size fragment was determined using a LabChip GX (PerkinElmer, Shelton, CT, USA) instrument.

#### 4.4.2. WGBS Libraries Construction 

gDNA was quantified using the Quant-iT™ PicoGreen^®^ dsDNA Assay Kit (Life Technologies). Libraries were generated with the NEBNext Ultra II DNA Library Prep Kit for Illumina (New England BioLabs) using 250 ng of input genomic DNA. Adapters were purchased from NEB. Size selection of libraries containing the desired insert size have been performed using sparQ PureMag Beads (Quantabio, Beverly, MA, USA). For whole genome bisilfite sequencing (WGBS), bisulfite conversion had been carried out with the EZ DNA Methylation-Lightning Kit (Zymo Research, Irvine, CA, USA). Libraries were quantified using the Kapa Illumina GA with Revised Primers-SYBR Fast Universal kit (Kapa Biosystems) and average size fragment was determined using a LabChip GX (PerkinElmer) instrument.

Data obtained were analyzed using several toolkits found in the methylKit package. WGBS allows for the investigation of genome-wide patterns of DNA methylation at a single base resolution. It involves the sodium bisulfite conversion of unmethylated cytosine to uracil. The remaining cytosine residues in the sequence represent the methylated cytosines in the genome, which is then mapped to a reference genome [59]. Binomial tests were applied and used to determine the observed methylation frequency against a bisulfite conversion reaction, and the percentage methylation (%methylation) levels were calculated at each base [60].

#### 4.4.3. WGS and WGBS Sequencing

The libraries were normalized and pooled and then denatured in 0.05 N NaOH and neutralized using HT1 buffer. ExAMP was added to the mix following the manufacturer’s instructions. The pool was loaded at 200 pM on an Illumina cBot and the flowcell was ran on a HiSeq X for 2 × 151 cycles (paired-end mode). A phiX library was used as a control and mixed with libraries at 1% level. The Illumina control software was HCS HD 3.4.0.38, the real-time analysis program was RTA v. 2.7.7. Program bcl2fastq2 v2.20 was then used to demultiplex samples and generate fastq reads. Bisulfite conversion rate was 99.7 and the spiked-in unmethylated lambda phage DNA was used to estimate the bisulfite conversion rate.

### 4.5. Bioinformatics Analysis of WGS Data

Adapter trimming was done by using Trim Galore software, version 0.6.7 with the “-q 30” option. Then reads were mapped to the Tair10 genome using the bwa-mem of BWA software 0.7.17 (r1188), and duplicates were marked by using the Picard tools software 3.1.1. Local realignments around SNPs and InDels were performed using GATK (a Genome Analysis Toolkit) which accounts for genome aligners, mapping errors and gives the consistent regions that contain SNPs and InDels. The resulting reads were quality controlled by Haplotype scores, and sample variant sites were called individually and jointly by using Haplotypercaller with GATK. The sites marked as a low-quality score by GATK were filtered out and used. The effects of variants in the genome sequences were classified using the SnpEff program [25]. Toolkits used included genomation to obtain a biological understanding of genomic intervals and the Functional Classification SuperViewer to create gene expression profiles and show the difference between samples. The genes nearest to the non-overlapping SNPs and InDels sites were annotated. 

### 4.6. Bioinformatics Analysis of WGBS Data

Raw sequencing reads were quality controlled and trimmed using Trim Galore software, version 0.6.7, similar to the WGB analysis initiation. The trimmed reads were subsequently aligned to the TAIR10 reference genome using the bisulfite mapping tool Bismark [61]. The methylated cytosines (Cs) were extracted from the aligned reads with the Bismark methylation extractor using default parameters followed by the computation of methylation frequency using the Bioconductor R package software, methylKit, version 1.28.0. The %methylation was calculated by counting the ratio of the frequency of Cs divided by reads with C or T at each base and computed at bases with coverage ≥ 10 [62].
%*Methylation* = (*Frequency of C* ÷ *read coverage*) × 100

Common bases covered across all samples were extracted and compared, and the differential hyper- and hypomethylated bases in each chromosome were extracted. The differentially methylated cytosines (DMCs) overlapping with genomic regions were assessed (in the preference for a promoter > exon > intron), and the average percentage methylation of DMCs around genes with distances of DMCs to the nearest transcription start sites (TSSs) were also calculated. Annotation analysis was performed with the genomation package within a methylKit to obtain a biological understanding of genomic intervals over the pre-defined functional regions like promoters, exons, and introns [63]. Functional commentary of the generated gene expression profiles was performed using the SuperViewer tool with Bootstrap to show the difference between samples [64]. Hierarchical clustering of samples was used to analyze for similarities and detect sample outliners based on the percentage methylation scores and a possible molecular signature. Also, Principal Component Analysis (PCA) was utilized for variations and any biological relevant clustering of samples. Scatterplots and bar plots showing the percentage of hyper-/hypo- methylated bases, overall chromosomes and heatmaps were used to visualize similarities and dissimilarities between DNA methylation profiles.

### 4.7. Analysis of Differentially Methylated Regions (DMRs)

DMRs information was analyzed over the predefined regions for all contexts; CpG (cytosine followed by guanine), CHG (cytosine followed by any nucleotide, followed by guanine), and CHH (cytosines followed by any nucleotides) on 100 bp and 1000 bp tiles across the genome to identify stochastic and treatment-associated DMRs [62]. The differential hyper-/hypo- methylated regions were also extracted and compared across samples. By default, DMRs were extracted with *q*-values < 0.01 and percent methylation difference > 25% to find out biologically relevant results; it was taken as arbitrary because > 50% would result near to nothing significant results. The differential methylation patterns between sample groups and methylation events of these differences per chromosome were extracted too. In summary, sliding windows of 100 bp and 1000 bp were considered for both DMRs and DMCs, and values were extracted based on at least 25% and 50% differences (*q*-values > 0.01) to assess significant differences among samples. 

### 4.8. Quality Control and Statistical Analysis of Sequencing Data

Mapped reads were obtained with a quality score of <30, differential hyper- and hypomethylated bases were extracted with *q*-values < 0.01 and percent methylation difference larger than 25% in methylKit. Heatmaps of differentially methylated bases were quantified at *q*-values < 0.01 and the percent methylation difference was more significant than 50%. The distances of DMCs to the nearest TSSs obtained from genomation were run at >25% and >50%. The TSSs distance to DMCs was extracted within +/−1000 bp and annotated at DMCs >50%. DNA methylation profiles obtained from melthylKit used the pairwise correlation coefficients of the percent methylation level and the 1-Pearson’s correlation distance for the hierarchical clustering of samples. Logistic regression and Fisher’s exact test were used to determine differential methylation with calculations of *q*-values and Benjamini-Hochberg for *p*-values corrections. The *t*-test for the mean difference between groups was calculated and extracted with *p*-values at least <0.05. Global genome methylation results were graphed using Microsoft Excel^®^ (MS) and output graphs from each corresponding program used.

## 5. Conclusions

Transgenerational memory of stress exposure is likely inherited through epigenetic mechanisms, including changes in DNA methylation, histone modifications and chromatin states, and it is likely in part triggered by non-coding RNAs [65]. Our genomic study revealed higher numbers of SNPs and InDels in the cold-stressed progenies. Similarly, epigenomic studies showed that the main differences occurred due to the hypomethylation at the non-symmetrical CHH context, likely indicating essential role of the RdDM machinery and ncRNAs in the process. The accumulation of the stress responsiveness over multiple generations due to the consecutive cold-stress could be guided and maintained by differential methylome, differential expression of ncRNAs and as a result by differential gene expression and changes in phenotype [66,67]. Baulcombe & Dean (2014) suggested that methylome variations in plants could be a potential cause of phenotypic differences to mitigate the environmental stresses [68]. 

It remains to be shown whether the progeny of cold-stressed plants have a different pattern of expression of ncRNAs and whether these ncRNAs map to the regions with differential methylation, especially at the CHH contexts. It would also be important to perform gene expression analysis and to correlate it to the methylation pattern. Finally, it would be good to demonstrate that the pattern of histone modifications also corresponds to the pattern of methylome and ncRNAome. Regardless, our study highlights that multigenerational exposure to cold stress has a substantial effect on the cold-stressed progenies in shaping their phenotypes, the genome and epigenome, which potentially suggests driving force of creating the variations across progeny generations. 

## Figures and Tables

**Figure 1 ijms-25-02795-f001:**
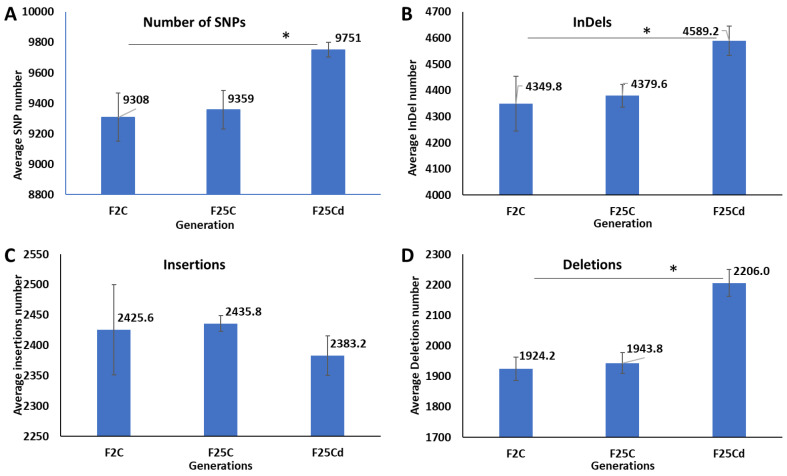
Comparison of SNPs and InDels in F2, F25C and F25Cd groups. A comparison of average SNPs (**A**), InDels (**B**), Insertions (**C**) and Deletions (**D**) variants in the genome. Y axis shows the average (with SD), calculated from five independent biological repeats. The asterisk above (*) shows a significant difference between the cold stressed progeny and parental or parallel control generations (*t*-test, two-sample assuming unequal variances; *p* < 0.05).

**Figure 2 ijms-25-02795-f002:**
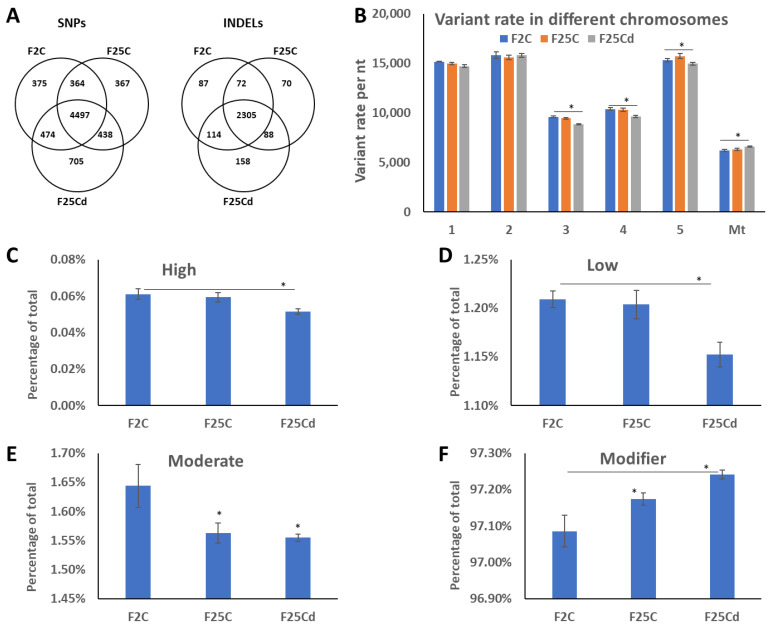
Analysis of SNPs uniqueness (**A**), SNPs distribution between chromosomes (**B**) and potential SNPs impact (**C**–**F**). (**A**) Venn diagrams of SNPs and InDels for F2C, F25C and F25Cd progenies. (**B**) Variant rate in different chromosomes shown as number of nucleotides per one SNP in each chromosome; Mt—mitochondrial genome. (**C**–**F**) Demonstrates the percentage of specific type of SNPs by their impact on gene expression, where “high” (**C**) has the highest impact on gene expression, protein composition or/and protein function, while “low” (**D**) and “moderate” (**E**) have less of an effect. “Modifier” (**F**) has the least effect on gene expression. All terms are from SnpEff [25]. Data are averaged (with SE) from five individual plants. Asterisks over the bar show significance (*t*-test, two-sample assuming unequal variances; *p* < 0.05) between F25Cd and either of F2C or F25Cd groups. In (**E**), asterisks show significant differences to F2C group. In (**F**), asterisk over F25C indicates significant (*p* < 0.05) difference to F2C group.

**Figure 3 ijms-25-02795-f003:**
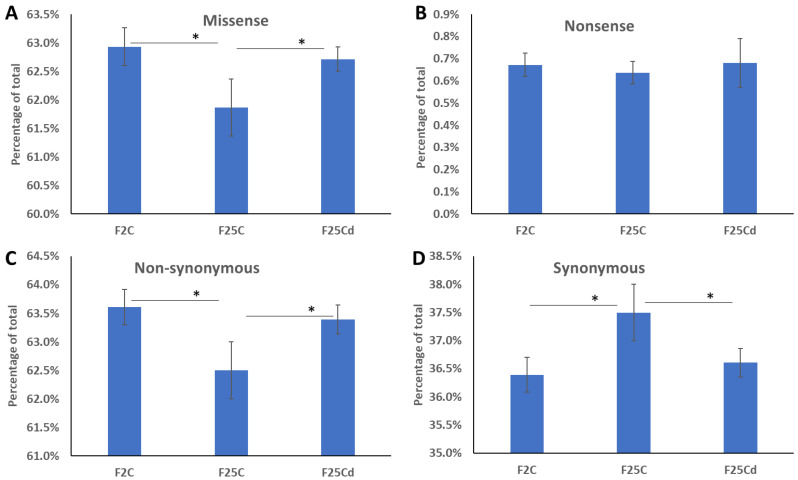
Percentage of missense (**A**), nonsense (**B**), non-synonymous (**C**) and synonymous (**D**) mutations in F2, F25C and F25Cd groups. Y axis shows the percentage of mutations of certain type, while X axis shows the group. Data are averaged (with SE) from five individual plants. Asterisks over the bar show significance (*t*-test, two-sample assuming unequal variances; *p* < 0.05) between groups.

**Figure 4 ijms-25-02795-f004:**
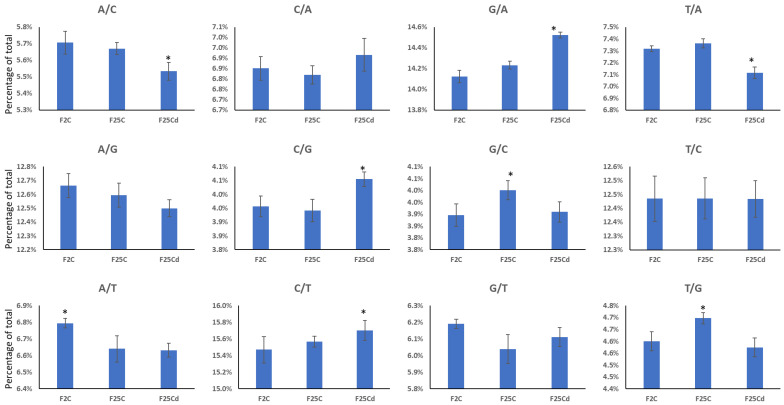
Type of nucleotide substitutions in F2, F25C and F25Cd groups. *Y* axis shows the percentage of mutations of certain type, while *X* axis shows the group. Data are averaged (with SE) from five individual plants. Asterisks over the bar show significance (*t*-test, two-sample assuming unequal variances; *p* < 0.05) between the indicated group and any of two other groups. A—adenine, C—cytosine, G—guanine, T—thymine.

**Figure 5 ijms-25-02795-f005:**
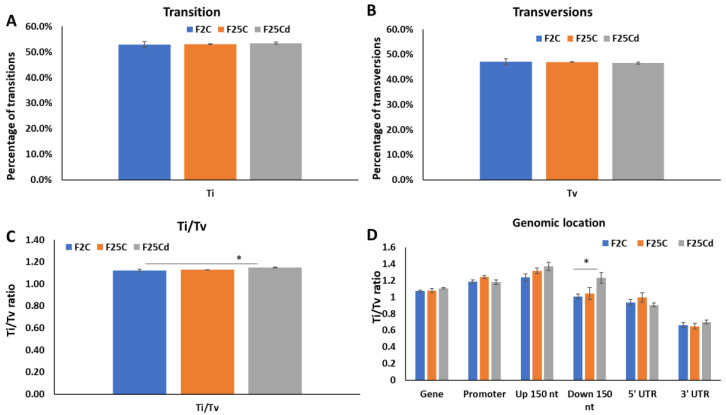
Percentage of transitions (**A**), transversions (**B**) and transition to transversion (Ti/Tv) ratio (**C**) in F2, F25C and F25Cd groups. Ti/Tv ratio in various genomic locations (**D**). Y axis shows either the percentage of Ti, Tv or Ti/Tv ratio, while X axis shows the group. Data are averaged (with SE) from five individual plants. Asterisks over the bar show significance (*t*-test, two-sample assuming unequal variances; *p* < 0.05) between the F25Cd and any of two other groups.

**Figure 6 ijms-25-02795-f006:**
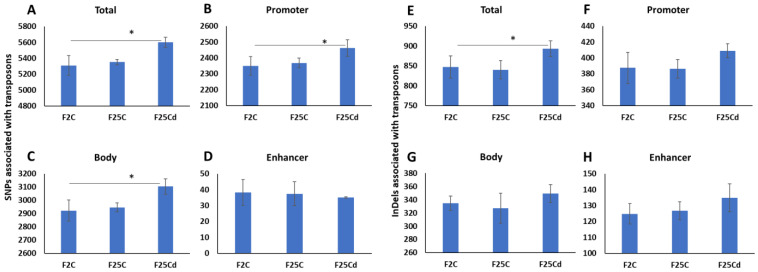
Number of SNPs (**A**–**D**) and InDels (**E**–**H**) associated with transposons. *Y* axis shows the number of SNPs or InDels associated with the entire transposon, promoter region, gene body region or enhancer element of the transposon. Data are averaged (with SD) from five individual plants. Asterisks over the bar show significance (*t*-test, two-sample assuming unequal variances; *p* < 0.05) between the F25Cd and any of two other groups.

**Figure 7 ijms-25-02795-f007:**
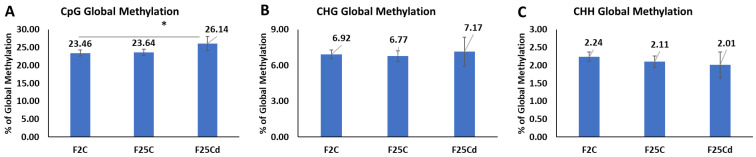
The average percentage (with SD, *n* = 5) of methylated cytosines in F2C, F25C, and F25Cd in the CpG (**A**) (cytosine followed by guanine), CHG (**B**) (cytosine followed by any nucleotide, followed by guanine), and CHH (**C**) (cytosines followed by any nucleotides) sequence contexts (H = A, T, C). F25Cd in the CpG region showed a statistically significant difference compared with the F2C or F25C. Asterisks over the bar show significance (*t*-test, two-sample assuming unequal variances; *p*-value < 0.05). Methylation levels were determined from reads with minimum coverage ≥ 10 mapped to TAIR 10 reference; data were analyzed by using Bismark.

**Figure 8 ijms-25-02795-f008:**
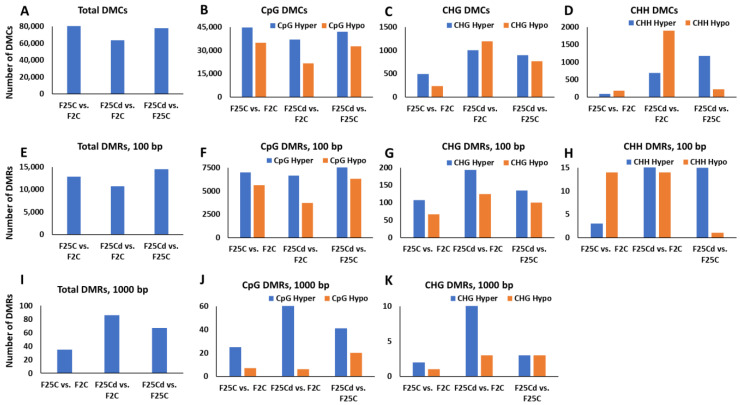
The number of differentially methylated cytosines (DMCs) and differentially methylated regions (DMRs) in F25H, F25C and F2C plants. (**A**) shows the total number of DMCs in F25H vs. F2C, F25H vs. F25C and F25C vs. F2C in all contexts, while (**B**–**D**) shows it in CG (**B**), CHG (**C**) or CHH (**D**) contexts. (**E**) shows the total number of DMRs in F25H vs. F2C, F25H vs. F25C and F25C vs. F2C in 100 bp window in all contexts, while (**F**–**H**) shows it in CG (**F**), CHG (**G**) or CHH (**H**) contexts. (**I**) shows the total number of DMRs in F25H vs. F2C, F25H vs. F25C and F25C vs. F2C in 1000 bp window in all contexts, while (**J**) shows it in CG and (**K**) in CHG contexts.

**Figure 9 ijms-25-02795-f009:**
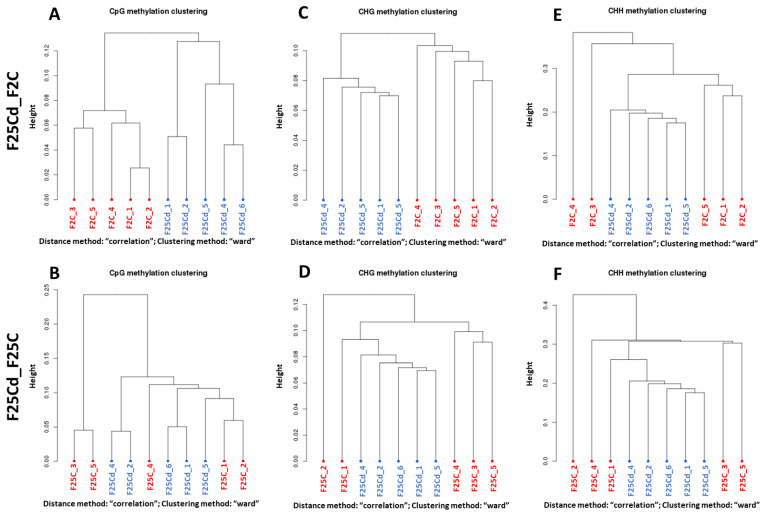
CpG (**A**,**B**), CHG (**C**,**D**) and CHH (**E**,**F**) methylation clustering of differentially methylated cytosines (DMCs) in the F25Cd vs. F2 (labelled as F25Cd_F2) (**A**,**C**,**E**) and F25Cd vs. F25C (labelled as F25Cd_F25C) (**B**,**D**,**F**) comparison groups. CpG—cytosine followed by guanine, CHG—cytosine followed by any nucleotide, followed by guanine, and CHH—cytosines followed by any nucleotides (H = A, T, C). “Height” indicates the distance of split. Hierarchical clustering of all fifteen methylomes was done by using Pearson’s correlation distance.

**Figure 10 ijms-25-02795-f010:**
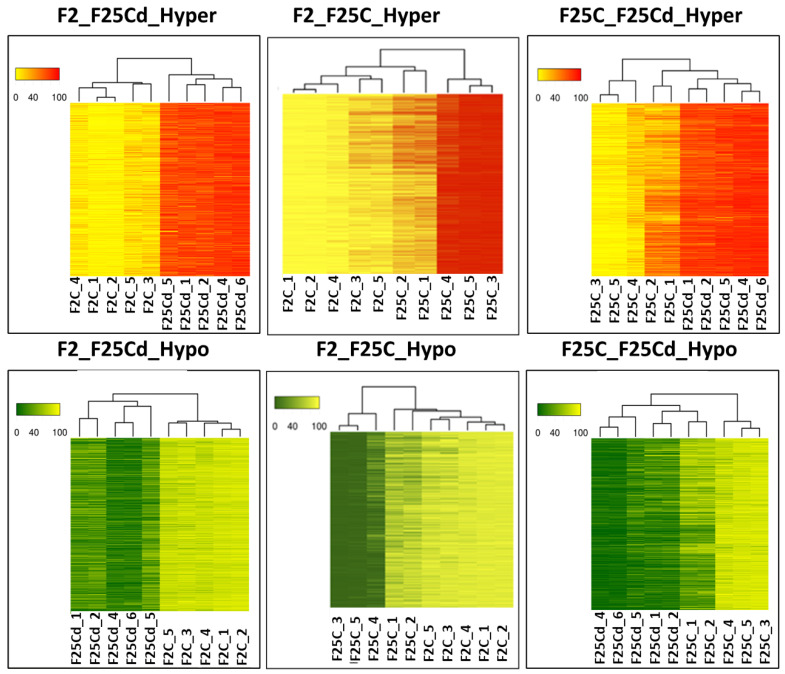
A hierarchical clustering heatmap analysis. Heat maps of DMCs for hypermethylated cytosines (**the upper panel**) and hypomethylated cytosines (**the lower panel**) in CG contexts in F25Cd vs. F2, F25C vs. F2C and F25Cd vs. F25C comparison groups, shown for positions with >50% difference (*q*-value < 0.01). “Hyper” indicates grouping of hypermethylated cytosines, while “hypo” indicates grouping of hypomethylated cytosines at CG sites. Color code indicate the methylation level from 0 to 100%. Height and width of dendrogram indicate the distance of split.

**Figure 11 ijms-25-02795-f011:**
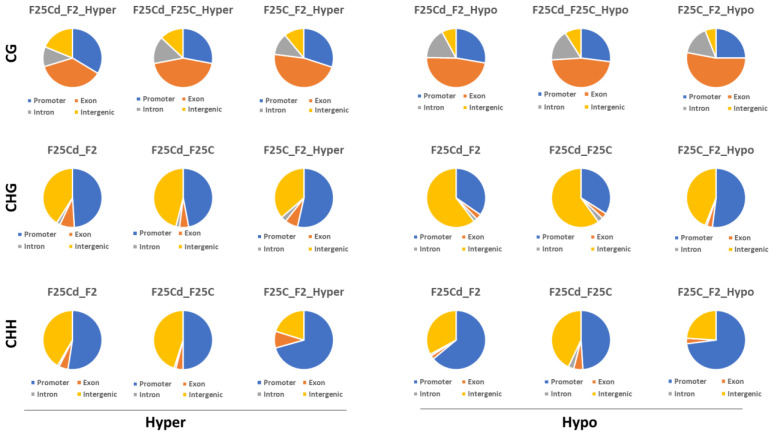
The percentages of differentially hypermethylated and hypomethylated DMCs in CG, CHG and CHH contexts in different genomic regions in F2C vs. F25Cd, F25C vs. F25Cd, and F2C vs. F25C comparison groups. The percentages plotted are the average (*n* = 5) percentages of DMCs overlapping various genomics regions, including promoters, exons, introns, and intergenic regions where DMCs were considered as regions with >25% difference in methylation with the coverage of at least 10 sequence reads per DMC.

**Figure 12 ijms-25-02795-f012:**
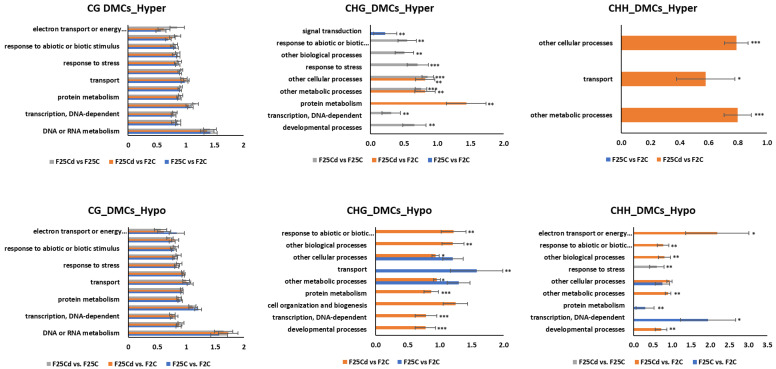
The enrichment analysis of hypermethylated DMCs in CG, CHG and CHH contexts and their classification based on the biological processes. The *Y*-axis shows significantly enriched biological processes. The *X*-axis is the normalized frequency with binomial coefficients as calculated by SuperViewer. Genes are classed with *p*-values < 0.05, ±bootstrap StdDev. Asterisks indicate the difference between and F25Cd vs. F25C or F25Cd vs. F2C and the control comparison F25C vs. F2C; one—*p* < 0.05, two—*p* < 0.01, three—*p* < 0.001 (*t*-test, two-sample assuming unequal variances).

**Table 1 ijms-25-02795-t001:** Functional annotation of SNPs.

Enrichment	F2C	F25C	F25Cd
**Biological process**			
transcription, DNA-dependent	1.3 ± 0.26	-	-
other metabolic processes	0.87 ± 0.08	0.82 ± 0.13	0.88 ± 0.08
other cellular processes	0.82 ± 0.08	0.79 ± 0.13	0.81 ± 0.08
cell organization and biogenesis	0.47 ± 0.16	-	0.51 ± 0.16
response to stress	0.69 ± 0.16	-	-
other biological processes	-	0.33 ± 0.19	-
protein metabolism	-	-	0.76 ± 0.17
**Molecular function**			
transcription factor activity	1.53 ± 0.33	-	-
protein binding	0.74 ± 0.16	-	0.74 ± 0.16
nucleotide binding	0.7 ± 0.19	0.13 ± 0.11 *	0.61 ± 0.19
other binding	-	0.73 ± 0.19	1.1 ± 0.12 *
transferase activity	-	0.32 ± 0.17	0.72 ± 0.19 *
hydrolase activity	-	0.23 ± 0.16	-
other enzyme activity	-	0.21 ± 0.14	-
**Cellular component**			
other membranes	0.8 ± 0.13	0.55 ± 0.17	0.85 ± 0.12
other cytoplasmic components	0.77 ± 0.11	0.53 ± 0.16	0.79 ± 0.12
chloroplast	0.72 ± 0.16	-	0.8 ± 0.16
other intracellular components	0.71 ± 0.12	0.64 ± 0.19	0.72 ± 0.13
cytosol	0.49 ± 0.20	-	-
plasma membrane	-	0.5 ± 0.23	-
nucleus	-	1.26 ± 0.17	-
other cellular components	-	-	0.46 ± 0.23
extracellular	-	-	1.38 ± 0.24
mitochondria	-	-	0.72 ± 0.18

An enrichment analysis of SNPs and associated genes and their classification based on biological processes, molecular function and cellular components. The normalized frequency (±bootstrap SD) with binomial coefficients was calculated by SuperViewer. Numbers above “1” show over-representation, while numbers below “1”—under-representation. Asterisks (*p* < 0.05) show significant difference from other groups. SNPs—single nucleotide polymorphism.

**Table 2 ijms-25-02795-t002:** Functional annotation of InDels.

Enrichment	F2C	F25C	F25Cd
**Biological process**			
signal transduction	1.23 ± 0.22	0.62 ± 0.18	1.20 ± 0.15
transport	1.20 ± 0.20	-	1.08 ± 0.14
transcription, DNA-dependent	1.12 ± 0.15	1.38 ± 0.2	1.38 ± 0.13
protein metabolism	1.10 ± 0.14	0.89 ± 0.13	0.91 ± 0.08
other metabolic processes	0.99 ± 0.06	1.00 ± 0.07	1.01 ± 0.04
other cellular processes	0.99 ± 0.05	0.93 ± 0.07	1.00 ± 0.04
developmental processes	0.87 ± 0.14	-	1.19 ± 0.12 *
response to abiotic or biotic stimulus	-	0.72 ± 0.13	1.09 ± 0.11 *
response to stress	-	-	1.25 ± 0.11
cell organization and biogenesis	-	-	0.85 ± 0.12
other biological processes	-	-	1.15 ± 0.13
**Molecular function**			
other molecular functions	1.54 ± 0.33	-	1.28 ± 0.23
nucleic acid binding	1.39 ± 0.24	1.36 ± 0.26	-
transporter activity	1.32 ± 0.28	-	-
transcription factor activity	1.22 ± 0.21	1.62 ± 0.26	1.53 ± 0.17
DNA or RNA binding	1.18 ± 0.13	1.19 ± 0.15	1.17 ± 0.09
protein binding	1.17 ± 0.12	-	1.11 ± 0.09
other binding	0.99 ± 0.07	0.87 ± 0.09	1.00 ± 0.07
transferase activity	0.89 ± 0.12	0.70 ± 0.13	0.90 ± 0.08
other enzyme activity	0.86 ± 0.08	0.89 ± 0.11	0.87 ± 0.07
hydrolase activity	-	0.81 ± 0.15	0.87 ± 0.09
nucleotide binding	-	0.58 ± 0.14	0.91 ± 0.10 *
kinase activity	-	-	1.15 ± 0.17
**Cellular component**			
other cellular components	1.22 ± 0.26	1.63 ± 0.31	1.34 ± 0.2
mitochondria	1.13 ± 0.13	-	1.08 ± 0.1
nucleus	1.07	1.03	1.03
other intracellular components	1.01 ± 0.1	0.87 ± 0.1	0.90 ± 0.07
other cytoplasmic components	0.94 ± 0.07	0.96 ± 0.08	0.97 ± 0.06
extracellular	0.84 ± 0.14	0.83 ± 0.14	1.05 ± 0.11
other membranes	0.83 ± 0.09	0.91 ± 0.1	0.89 ± 0.06
chloroplast	0.78 ± 0.11	0.76 ± 0.12	0.83 ± 0.09
Golgi apparatus	0.66 ± 0.19	-	0.67 ± 0.15
plasma membrane	-	0.77 ± 0.13	0.87 ± 0.08
cytosol	-	0.56 ± 0.17	0.85 ± 0.11 *
plastid	-	-	0.79 ± 0.14
cell wall	-	-	1.27 ± 0.23

An enrichment analysis of InDels (insertions and deletions) and associated genes and their classification based on biological processes, molecular function and cellular components. The normalized frequency (±bootstrap SD) with binomial coefficients was calculated by SuperViewer. Numbers above “1” show over-representation, while numbers below “1”—under-representation. Asterisks (*p* < 0.05) show significant difference from other groups.

## Data Availability

Original sequencing data were deposited at https://www.ncbi.nlm.nih.gov/bioproject/PRJNA802915; accessed on 17 December 2023.

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
