# Peer review of "Genomic and Epigenomic Changes in the Progeny of Cold-Stressed Arabidopsis thaliana Plants"

_ijms, 2024, doi:10.3390/ijms25052795_

Round 1

Reviewer 1 Report

Comments and Suggestions for Authors

 Dear authors,

This article concerns “ Genomic and epigenomic changes in the progeny of cold- stressed Arabidopsis thaliana plants, by Ashif Rahman, Narendra Yadav, Boseon Byeon, Yaroslav Ilnytskyy, and Igor Kovalchuk. As interesting data (apparently directly extracted from master degree text (https://opus.uleth.ca/server/api/core/bitstreams/dd28e230-bf8d-4653-9e64-42748882d78b/content, and in https://www.proquest.com/openview/66cc2931392afff762857042425cb04c/1?pq-origsite=gscholar&cbl=18750&diss=y ) visible in the internet site of Department of Biological Sciences University of Lethbridge Lethbridge, Alberta, Canada, not cited in the present submission), with especially rarely documented morphological phenotypes of Arabidopsis, I recommend it for an international audience in this journal, however several points have to be considered by the authors, and a major revision is requested.

Please notice that in order to bring a broad audience to this article and to this journal, for specialists and non-specialists, the five major points of my comments (at the beginning) are very important (mandatory…) for a suitable value of the article. Minor points are also enhanced at the end of this review.

I deeply hope to see this good article published soon,

The five major points are:

1-1     As a botanist I am, the first point embarrassing me a lot concerns the morphological phenotypes, see my four points just after (1/2/3/4/) which need clarification(s). 1/ First of all, as it is rarely studied in molecular biology, figure S1 has to be inserted in the main text, not in supplementary materials; moreover, provide detailed photos of leaves, bolting and seeds (with scale bar and its value just above the scale to be read rapidly), as it concerns your figure 1; although interesting, the present photos are not receivable. 2/ For figure 1 indicate precisely ("approximately" has no statistical meaning) the number of plants (leaves, seeds etc…) measured, 30 being the minimum required for significant statistics. 3/ In figure 1 too, You should provide each diagram with comparisons of significant differences or not (= tendencies) between all generations (this is quite easy to provide for instance with R using clearly p values) instead of significance within one generation as in the present figure, if I understood well; it will be much more clear for the discussion phenotypes part; moreover for instance I do not understand the difference between F10C no stress (on the left side) and F10C cold stress (on the right side), what does exactly mean "control"? What is for instance F10CD if it appears also in "no stress" left part? 4/ About the data provided in figure 1, although they are used in cited former references, leaf number (apparently not so stable according to your interesting point 3.1.) should be related with leaf size (= surface, directly related with photosynthesis and energy, much more relevant in many plants, the best criteria being stomatal density and activity or not of the stomata) which seems quite evident with the diversity of photos provided in figure 1s. For flowering data, the number of flowers seems also relevant according to the photos. For seeds, their weight is usually much more relevant in plants, as directly related with the quantity of molecules (= activity of genes, which is your topics); seed height means actually seed length? Weight of plants appears in the material and methods but not in figure 1or in the text (?). For bolting, the numbers of days for bolting (instead of a strict barrier at 31 days of bolting) seems much more relevant, also the number of days for flowering instead of 31-42. Finally the first paragraph of 3.1. is not understandable: the first sentence is not represented in figure 1(?); what is the relationship between slow germination and leaf numbers?; what is the relationship between the age of seeds and leaf numbers? The last sentence is not clear at all.

2-2     In the introduction and discussion, terms and concepts are used far too strongly and more or less apparently mixed, making this text not clear at all. Please, in a first step, give some precise definitions of your mostly used words (epigenesis-mutation (the differences or affinities between these two words is not clear at all), transgeneration, microevolution, adaptation, speciation), then clarify the taxa (or subtaxa like lines or accessions very frequent in Arabidopsis) used for these studies and limit your (promising) purpose just for these taxa. For instance: in the introduction, "mostly disappear when stress conditions are no longer present" means "accommodation-plasticity", which is very different from your purpose (?); in the introduction too, "epigenetic variants" are, so far known, very different from  "adaptation, microevolution and potentially speciation" as you pretend (?); in the introduction too, "short term" is a little bit conflicting (contradictory?) with microevolution (?).

3-3     I do not see clearly the relationship between the phenotypes observed and the putative function(s) of the genes as enumerated for instance in tables 1 and 2; most of the time correlation(s) are shown in experiments, there are (so) many correlations in plants…; function is a very complex concept related with “chains of causation”, which need detailed physiology and anatomy and molecular (and etc…) studies, usually a very hard work!; in this sense the phenotype part is not clearly connected to the genes sensu lato part (with so diverse functions); moreover, in the introduction and discussion, please precise the taxa (subtaxa…) involved in your cited references as generalizations are far too strong, especially for this kind of topics.

4-4     Concerning the rates of mutations and epigenesis provided in this study, these rates should be compared with rates generally observed in Arabidopsis, also in other taxa (subtaxa) as I am surprised that this Arabidopsis line may show some changes so rapidly. This remark is linked with references (please check for more) enhancing random (or not, or more or less) epigenesis and/or mutation(s). For instance, in 5.1, how long is a generation = 1 year of duration or shorter? Since it seems to be very rapid (as experimental) compared with wild Arabidopsis, how may it influence changes sensu lato of genes material for your Arabidopsis line? In 3.2.1., since there is apparently no selection in your seeds over generations, how can you write that "It is possible that there exist some sort of purifying selection in the progeny of cold-stressed plants."?

5-     References already taken in account by the authors are of interest, however checking briefly in word of science WOS and scilit (from mdpi) with some key-words of this manuscript (Arabidopsis, epigenome, mutation, phenotype, cold stress…) other references appear (especially recent ones), and they should be updated and used (if relevant…) in order to provide a larger view of these interesting researches. Among these are the followings (check all others)

[1-23]

1.         Cahn, J.; Lloyd, J.P.B.; Karemaker, I.D.; Jansen, P.W.T.C.; Pflueger, J.; Duncan, O.; Petereit, J.; Bogdanovic, O.; Millar, A.H.; Vermeulen, M.; et al. Characterization of DNA methylation reader proteins of Arabidopsis thaliana; 2023.

2.         Casati, P.; Roldán-Arjona, T.; Molinier, J. Editorial: Plant Genome-Epigenome Integrity Under Environmental Stress. Frontiers in Plant Science 2020, 11.

3.         Charlesworth, D. Faculty Opinions recommendation of Mutation bias reflects natural selection in Arabidopsis thaliana. 2022.

4.         Denkena, J.; Johannes, F.; Colomé-Tatché, M. Region-level epimutation rates in Arabidopsis thaliana. Heredity 2021, 127.

5.         Guangchao, L.; Liu, F.-X.; Wang, Y.; Liu, X. A Novel Long Non-Coding RNA CIL1 Enhances Cold Stress Tolerance in Arabidopsis; 2022.

6.         Kerbler, S.M.; Wigge, P.A. Temperature Sensing in Plants. Annual Review of Plant Biology 2023, 74.

7.         Lim, C.W.; Lee, S.C. Arabidopsis SnRK2.3/SRK2I plays a positive role in seed germination under cold stress conditions. Environ Exp Bot 2023, 212.

8.         Liu, G.; Liu, F.; Wang, Y.; Liu, X. A novel long noncoding RNA CIL1 enhances cold stress tolerance in Arabidopsis. Plant Science 2022, 323.

9.         Liu, W.; Wang, T.; Wang, Y.; Liang, X.; Han, J.; Han, D. MbMYBC1, a M. baccata MYB transcription factor, contribute to cold and drought stress tolerance in transgenic Arabidopsis. Frontiers in Plant Science 2023, 14.

10.       Maumus, F.; Quesneville, H. Ancestral repeats have shaped epigenome and genome composition for millions of years in Arabidopsis thaliana. Nat Commun 2014, 5.

11.       Maurya, B.; Sharma, L.; Kumari, S.; Rai, N.; Pandey-Rai, S. Plant Ecogenomics: Challenges and; 2022.

12.       Monroe, J.G.; Murray, K.D.; Xian, W.; Carbonell-Bejerano, P.; Fenster, C.B.; Weigel, D. Report of mutation biases mirroring selection in Arabidopsis thaliana unlikely to be entirely due to variant calling errors; 2022.

13.       Monroe, J.G.; Srikant, T.; Carbonell-Bejerano, P.; Becker, C.; Lensink, M.; Exposito-Alonso, M.; Klein, M.; Hildebrandt, J.; Neumann, M.; Kliebenstein, D.; et al. Mutation bias reflects natural selection in Arabidopsis thaliana. Nature 2022, 602.

14.       Monroe, J.G.; Srikant, T.; Carbonell-Bejerano, P.; Becker, C.; Lensink, M.; Exposito-Alonso, M.; Klein, M.; Hildebrandt, J.; Neumann, M.; Kliebenstein, D.; et al. Author Correction: Mutation bias reflects natural selection in Arabidopsis thaliana. Nature 2023, 620.

15.       Monroe, J.G.; Srikant, T.; Carbonell-Bejerano, P.; Exposito-Alonso, M.; Weng, M.-L.; Rutter, M.T.; Fenster, C.B.; Weigel, D. Mutation bias shapes gene evolution in Arabidopsis thaliana; 2020.

16.       Petegrosso, R.; Song, T.; Kuang, R. Hierarchical Canonical Correlation Analysis Reveals Phenotype, Genotype, and Geoclimate Associations in Plants. Plant Phenomics 2020, 2020.

17.       Raju, S.K.K.; Lensink, M.; Kliebenstein, D.J.; Niederhuth, C.; Monroe, G. Epigenomic divergence correlates with sequence polymorphism in Arabidopsis paralogs. New Phytologist 2023, 240.

18.       Stassen, J.H.M.; López, A.; Jain, R.; Pascual-Pardo, D.; Luna, E.; Smith, L.M.; Ton, J. The relationship between transgenerational acquired resistance and global DNA methylation in Arabidopsis. Sci Rep-Uk 2018, 8.

19.       Veitia, R.A. Who ever thought genetic mutations were random? Trends Plant Sci 2022, 27.

20.       Virdi, K.S.; Laurie, J.D.; Xu, Y.-Z.; Yu, J.; Shao, M.-R.; Sanchez, R.; Kundariya, H.; Wang, D.; Riethoven, J.-J.M.; Wamboldt, Y.; et al. Arabidopsis MSH1 mutation alters the epigenome and produces heritable changes in plant growth. Nat Commun 2015, 6.

21.       Wang, X.; Zhang, X.; Song, C.-P.; Gong, Z.; Yang, S.; Ding, Y. PUB25 and PUB26 dynamically modulate ICE1 stability via differential ubiquitination during cold stress in Arabidopsis. THE PLANT CELL ONLINE 2023, 35.

22.       Yan, Y.; Li, Z.; Li, Y.; Wu, Z.; Yang, R. Correlated Evolution of Large DNA Fragments in the 3D Genome of Arabidopsis thaliana. Molecular Biology and Evolution 2020, 37.

23.       Zhang, Y.; Wu, L.; Liu, L.; Jia, B.; Ye, Z.; Tang, X.; Heng, W.; Liu, L. Functional Analysis of PbbZIP11 Transcription Factor in Response to Cold Stress in Arabidopsis and Pear. Plants 2023, 13.

 Minor points are

1 As I am involved in plant taxonomy I am very sensible to correct plant taxa names which make their homogeneity and precision at the international level. So write all plants with full  latin names in italics, followed by the names of the author(s) (not in italics) at least the first time they appear in the text (from the beginning of the introduction); see carefully all parts (including tables and figures) of your text as there are numerous mistakes. Use international Plant Names Index (IPNI) https://www.ipni.org/, or equivalent. For instance, just for cultivars (or lines, accessions…if there is no author’s name(s), put the reference where this name appears firstly in the literature. Then once it is written correctly in full letters you can write further in the text only the initial (for instance A. for Arabidopsis, plus species name (Arabidopsis alone has no meaning…)).

2 Restrict the results part to only results (values etc.) and remove all remarks and sentences which belong actually to the discussion part (for instance, just in  in 2.2.2 "This analysis confirms...", "may suggest...",   "it can be suggested"…; see for all other parts of the results. Moreover, in the discussion the sentence "This study aimed..." should be in the conclusion. Some results do not appear in the discussion (use precise values…) while results and discussion should be linked much more.

3 I am a little bit doubtful about some key-words like "phenotypic resilience” (“resilience” is nowadays more and more used for so many purposes...).

4 In the introduction, "rapid climate changes" is not fully accepted by many ecologists.

5 For figure 1.A, write “control progeny” instead of “cotrol”.

6 For figure 5, in order to be read rapidly indicate in the caption in full letters the abbreviations of the nucleotides.

7 For figure 6, indicate more clearly the meaning of Ti and Tv.

8 For table 1, indicate somewhere in full letters the meaning of SNP.

9 For table 2, indicate somewhere in full letters the meaning of INDEL.

10 For figure 8, indicate in full letters the meaning of CpG, CHG, CHH.

11 The captions of figure 9-11 contain too many abbreviations making this part very difficult to understand rapidly.

12 In the discussion, although it is used in many recent articles, the word "sessile" is not appropriate" in this case as it has another very different meaning in botany;

13 For figure S1, in the caption put "left" instead of "let"?

14 For figure S2, F2 is F1C?

15 For figure S4, indicate in the caption the mean of abbreviations like DMC CH, CHG, CHH; do the same for figure S5, S6, S7.

16 In 3.1., "the being less sensitive to the environmental conditions" is not understandable (what means actually sensitive?).

17 At the end of 3.1., what is the use of the last sentence "on the other hand..." in this study?

18 At the end of 3.3.2, precise the taxa involved in the references 63 and 64.

19 In 3.3.5., precise in full letters the meaning of CG, CHG and CHH.

20 For figure 1, indicate in full letters the meaning of dpg.

Author Response

Reviewer #1

Please notice that in order to bring a broad audience to this article and to this journal, for specialists and non-specialists, the five major points of my comments (at the beginning) are very important (mandatory…) for a suitable value of the article. Minor points are also enhanced at the end of this review.  I deeply hope to see this good article published soon. 

The five major points are:

  • As a botanist I am, the first point embarrassing me a lot concerns the morphological phenotypes, see my four points just after (1/2/3/4/) which need clarification(s). 1/ First of all, as it is rarely studied in molecular biology, figure S1 has to be inserted in the main text, not in supplementary materials; moreover, provide detailed photos of leaves, bolting and seeds (with scale bar and its value just above the scale to be read rapidly), as it concerns your figure 1; although interesting, the present photos are not receivable.

Our response:

The main reason we put this figure into the Supplementary materials was that we do not have any better figures. The research was done in 2019, and the student has left a while ago. This was all I could get from him. Meanwhile, we deleted all phenotypic data entirely.

2/ For figure 1 indicate precisely ("approximately" has no statistical meaning) the number of plants (leaves, seeds etc…) measured, 30 being the minimum required for significant statistics.

Our response:

The figure was deleted.

3/ In figure 1 too, You should provide each diagram with comparisons of significant differences or not (= tendencies) between all generations (this is quite easy to provide for instance with R using clearly p values) instead of significance within one generation as in the present figure, if I understood well; it will be much more clear for the discussion phenotypes part; moreover for instance I do not understand the difference between F10C no stress (on the left side) and F10C cold stress (on the right side), what does exactly mean "control"? What is for instance F10CD if it appears also in "no stress" left part?

4/ About the data provided in figure 1, although they are used in cited former references, leaf number (apparently not so stable according to your interesting point 3.1.) should be related with leaf size (= surface, directly related with photosynthesis and energy, much more relevant in many plants, the best criteria being stomatal density and activity or not of the stomata) which seems quite evident with the diversity of photos provided in figure 1s. For flowering data, the number of flowers seems also relevant according to the photos. For seeds, their weight is usually much more relevant in plants, as directly related with the quantity of molecules (= activity of genes, which is your topics); seed height means actually seed length? Weight of plants appears in the material and methods but not in figure 1or in the text (?). For bolting, the numbers of days for bolting (instead of a strict barrier at 31 days of bolting) seems much more relevant, also the number of days for flowering instead of 31-42. Finally the first paragraph of 3.1. is not understandable: the first sentence is not represented in figure 1(?); what is the relationship between slow germination and leaf numbers?; what is the relationship between the age of seeds and leaf numbers? The last sentence is not clear at all.

Our response:

We deleted phenotypic data. We decided to focus on omics data.

2-2     In the introduction and discussion, terms and concepts are used far too strongly and more or less apparently mixed, making this text not clear at all. Please, in a first step, give some precise definitions of your mostly used words (epigenesis-mutation (the differences or affinities between these two words is not clear at all), transgeneration, microevolution, adaptation, speciation), then clarify the taxa (or subtaxa like lines or accessions very frequent in Arabidopsis) used for these studies and limit your (promising) purpose just for these taxa. For instance: in the introduction, "mostly disappear when stress conditions are no longer present" means "accommodation-plasticity", which is very different from your purpose (?); in the introduction too, "epigenetic variants" are, so far known, very different from  "adaptation, microevolution and potentially speciation" as you pretend (?); in the introduction too, "short term" is a little bit conflicting (contradictory?) with microevolution (?).

 Our response:

We have made changes in the introduction.

3-3     I do not see clearly the relationship between the phenotypes observed and the putative function(s) of the genes as enumerated for instance in tables 1 and 2; most of the time correlation(s) are shown in experiments, there are (so) many correlations in plants…; function is a very complex concept related with “chains of causation”, which need detailed physiology and anatomy and molecular (and etc…) studies, usually a very hard work!; in this sense the phenotype part is not clearly connected to the genes sensu lato part (with so diverse functions); moreover, in the introduction and discussion, please precise the taxa (subtaxa…) involved in your cited references as generalizations are far too strong, especially for this kind of topics.

Our response:

I am not sure I understand the questions/concerns here. Table 1 lists functional annotation of SNPs as calculated by SuperViewer.

4-4     Concerning the rates of mutations and epigenesis provided in this study, these rates should be compared with rates generally observed in Arabidopsis, also in other taxa (subtaxa) as I am surprised that this Arabidopsis line may show some changes so rapidly. This remark is linked with references (please check for more) enhancing random (or not, or more or less) epigenesis and/or mutation(s). For instance, in 5.1, how long is a generation = 1 year of duration or shorter? Since it seems to be very rapid (as experimental) compared with wild Arabidopsis, how may it influence changes sensu lato of genes material for your Arabidopsis line? In 3.2.1., since there is apparently no selection in your seeds over generations, how can you write that "It is possible that there exist some sort of purifying selection in the progeny of cold-stressed plants."?

Our response:

Comparison of mutation rate with other studies would be very difficult as in our case, it was a cumulative rate over 25 generations. We did not calculate the rate per generation, so we can not compare it with others – most report mutation rate per nt per generation. This would be possibly if we analyzed generations F2 with F3 or F24 with F25. We do not think that this line has specifically high level of mutations, as it is a line (Columbia) that we and others (Ries, Molinier, Pecinka, Hohn etc.) used for many years.

5-     References already taken in account by the authors are of interest, however checking briefly in word of science WOS and scilit (from mdpi) with some key-words of this manuscript (Arabidopsis, epigenome, mutation, phenotype, cold stress…) other references appear (especially recent ones), and they should be updated and used (if relevant…) in order to provide a larger view of these interesting researches. Among these are the followings (check all others)

 Our response:

Thanks, we checked the references and used some of them.

 Minor points are

1 As I am involved in plant taxonomy I am very sensible to correct plant taxa names which make their homogeneity and precision at the international level. So write all plants with full  latin names in italics, followed by the names of the author(s) (not in italics) at least the first time they appear in the text (from the beginning of the introduction); see carefully all parts (including tables and figures) of your text as there are numerous mistakes. Use international Plant Names Index (IPNI) https://www.ipni.org/, or equivalent. For instance, just for cultivars (or lines, accessions…if there is no author’s name(s), put the reference where this name appears firstly in the literature. Then once it is written correctly in full letters you can write further in the text only the initial (for instance A. for Arabidopsis, plus species name (Arabidopsis alone has no meaning…)).

Our response:

We only used Arabidopsis thaliana; even so it is very common to write Arabidopsis, like tomato, corn, wheat etc., we have introduced the whole name in most cases.

2 Restrict the results part to only results (values etc.) and remove all remarks and sentences which belong actually to the discussion part (for instance, just in  in 2.2.2 "This analysis confirms...", "may suggest...",   "it can be suggested"…; see for all other parts of the results. Moreover, in the discussion the sentence "This study aimed..." should be in the conclusion. Some results do not appear in the discussion (use precise values…) while results and discussion should be linked much more.

Our response:

We have removed it in most of the cases. At the same time, I would like to say that it is a matter of preference and style – many comment data in the results sections.

3 I am a little bit doubtful about some key-words like "phenotypic resilience” (“resilience” is nowadays more and more used for so many purposes...).

Our response:

We deleted phenotypic data.

4 In the introduction, "rapid climate changes" is not fully accepted by many ecologists.

Our response:

Noted and corrected. Again, some reviewers would want me to put “rapid”…

5 For figure 1.A, write “control progeny” instead of “cotrol”.

 Our response:

We deleted phenotypic data.

6 For figure 5, in order to be read rapidly indicate in the caption in full letters the abbreviations of the nucleotides.

Our response:

Done.

7 For figure 6, indicate more clearly the meaning of Ti and Tv.

Our response:

Done

8 For table 1, indicate somewhere in full letters the meaning of SNP.

Our response:

Done

9 For table 2, indicate somewhere in full letters the meaning of INDEL.

Our response:

Done

10 For figure 8, indicate in full letters the meaning of CpG, CHG, CHH.

Our response:

Done

11 The captions of figure 9-11 contain too many abbreviations making this part very difficult to understand rapidly.

Our response:

I am not sure what can be done about it.

12 In the discussion, although it is used in many recent articles, the word "sessile" is not appropriate" in this case as it has another very different meaning in botany;

Our response:

We replaced it.

13 For figure S1, in the caption put "left" instead of "let"?

Our response:

Done.

14 For figure S2, F2 is F1C?

Our response:

Corrected.

15 For figure S4, indicate in the caption the mean of abbreviations like DMC CH, CHG, CHH; do the same for figure S5, S6, S7.

Our response:

Done.

16 In 3.1., "the being less sensitive to the environmental conditions" is not understandable (what means actually sensitive?).

Our response:

Deleted

17 At the end of 3.1., what is the use of the last sentence "on the other hand..." in this study?

 Our response:

Deleted

18 At the end of 3.3.2, precise the taxa involved in the references 63 and 64.

Our response:

Done.

19 In 3.3.5., precise in full letters the meaning of CG, CHG and CHH.

Our response:

This is described in Methods and Figure captions.

20 For figure 1, indicate in full letters the meaning of dpg.

Our response:

This is now deleted.

Reviewer 2 Report

Comments and Suggestions for Authors

The authors studied the Arabidopsis thaliana plants exposed to cold stress for 25 continuous generations. It is therefore a large study. They did so to analyse the genomic and epigenomic changes in the progeny of cold-stressed Arabidopsis thaliana plants. More precisely, they have stressed Arabidopsis plants with cold for 25 generations; they analyzed phenotypic, genetic and epigenetic changes and response to stress in the first and 25th generations of plants grown in normal conditions and cold-stressed.

They found the progeny of cold-stressed plants exhibited the higher rate of missense non-synonymous mutations as compared to the progeny of control plants. They link that with epigenetic changes that were more common in the CHG and CHH contexts and favored hypomethylation.

It is an interesting study. I have nonetheless concerns. The first is about the number of generations used. 25 generations is a lot. It would have really been interesting to have a dynamics analysis throughout the 25 generation and not only the final ones. This would be necessary to understand how quick the cold stress does trigger these genetic responses. Besides, are they maintained in generations grown without stress (experiments where plants are submitted to stresses but the progeny are not).

I encourage the authors to do such experiments in their future work.

It is a pity there is no gene response to see if these epigenetic changes have an impact on expression of the COR genes.

 I have big concerns about the way results are presented and the stats done (or the way the results are represented). See below.

Concerning this paper, the authors start with “Analysis of the leaf number showed that all groups had more leaves than F1 parental control (p<0.05) (Figure 1);”

They do not even explain the experimental set up nor do they explain the naming of the groups. They do it lines 739-750 but this is very far from the first line of the result section. So it is very important authors write and explain the set up in few sentences ate the beginning of the Result section.

They use F25C and F25Cd naming. Would it be possible to write F25Cold instead of F25Cd? It is just 2 more letters and it makes everything clearer? so F25C for control line and F25Cold for cold line.

Please make all the changes trough the text.

It is very difficult to understand figure1; You do not explain in the text what is cold stress and No stress? When is it applied? for how long??

line 101: fewer.

line 171: what test is done?

Figure. 3: is F2 F2C like in figure 2 and like in Fig3A? F2 has no meaning.

You introduce the comparison with F2C without clearly explaining/justifying it…. Why comparing with F2C? What about F2Cold? Because F2 is supposedly the most representative of the initial population?

Figure 3B: what is “Mt”?

In figure caption, can you indicate what test is done to decipher if it is significant or not?

in figure 3 caption you write “Asterisks over the bar show significance (p<0.05) between groups.” not clear for instance in 3B asterisks show significant differences between what and what??:  would noty I be possible to have anova and letters?

in 3B asterisks indicates difference between F25Cold and F2C? same for 3

but what about 3E and 3F??

In figure 4 the horizontal lines have not the same width… (in 4A and 4C).

Figures 4, 5, 6: what about F2? is that F2C??

Figure 5;  here again the asterisk is supposed to indicate significant differences between the indicated group and two other groups. By default, the other groups are not significantly different? It would be better to have letters or clear horizontal line indicating the pairwise comparison.

in figure 6 you indicate “Asterisks over the bar show significance (p<0.05) between the indicated group and two other groups.”  but the horizontal line would suggest a comparison between F25Cold and F2C… it is confusing.

same remark for fig7

Why is figure 7 divided in two rectangles??

Figure 10. write F25Cd vs. F2 and not F25Cd _F2. same for F25Cold vs. F25C. I note that you write F2_ct and not F2C for methylomes…..Here again another nomenclature….

line 391 you write “In contrast, in the F25Cd vs. F25C group comparison, F25Cd samples were

not as clearly separated from F25C group.” I do not think this is true of Fig11D and 11F.

line 389-391: indicate figure panels readers are supposed to look at.

Figure 12/ rewrite the titles of each anel. Do not use “_” for versus.

why for F25_CD you have F25_Cd6 but not F25_Cd3?

in some panels you have numbered 1 to 5 and in other you have numbered your samples 1 to 10 (even though they do not correspond to the same lines)!! keep 1 to 5.

I have small remarks to make:

line 88-89: is that correct? I think you did also the analysis at generations 10, 15 and 20? Am I right?

You wrote “Since they can not escape stress, they have to develop mechanisms of remembering stress exposures somatically and passing it to the progeny.”

write « . They have to developed mechanisms of remembering stress exposures somatically and passing it to the progeny.”

I do not agree with the fact that plants “had to”…

line 11 Arabidopsis thaliana in italics

line 18: maybe you need to explain what are CHG and CHH contexts for readers not at ease with genetics. What does H stand for?

Comments on the Quality of English Language

quite OK

Author Response

Reviewer #2

It is an interesting study. I have nonetheless concerns. The first is about the number of generations used. 25 generations is a lot. It would have really been interesting to have a dynamics analysis throughout the 25 generation and not only the final ones. This would be necessary to understand how quick the cold stress does trigger these genetic responses. Besides, are they maintained in generations grown without stress (experiments where plants are submitted to stresses but the progeny are not).

I encourage the authors to do such experiments in their future work.

Our response:

Yes, this is planned.

It is a pity there is no gene response to see if these epigenetic changes have an impact on expression of the COR genes.

Our response:

Again, this is planned.

Concerning this paper, the authors start with “Analysis of the leaf number showed that all groups had more leaves than F1 parental control (p<0.05) (Figure 1);”

They do not even explain the experimental set up nor do they explain the naming of the groups. They do it lines 739-750 but this is very far from the first line of the result section. So it is very important authors write and explain the set up in few sentences ate the beginning of the Result section.

Our response:

We have deleted phenotypic data.

They use F25C and F25Cd naming. Would it be possible to write F25Cold instead of F25Cd? It is just 2 more letters and it makes everything clearer? so F25C for control line and F25Cold for cold line.

Our response:

F25Cold distorts most figures because of the longer word.

It is very difficult to understand figure1; You do not explain in the text what is cold stress and No stress? When is it applied? for how long??

Our response:

Deleted.

line 101: fewer.

Our response:

Corrected.

line 171: what test is done?

Our response:

t-test, two-sample assuming unequal variances

Figure. 3: is F2 F2C like in figure 2 and like in Fig3A? F2 has no meaning.

Our response:

Corrected.

You introduce the comparison with F2C without clearly explaining/justifying it…. Why comparing with F2C? What about F2Cold? Because F2 is supposedly the most representative of the initial population?

Our response:

We wrote in the methods that we ran out of F1 seeds and then they were not viable; so F2C was the “earliest” control generation we had.

Figure 3B: what is “Mt”?

 Our response:

Mitochondrial – the information was added.

In figure caption, can you indicate what test is done to decipher if it is significant or not?

Our response:

Done.

in figure 3 caption you write “Asterisks over the bar show significance (p<0.05) between groups.” not clear for instance in 3B asterisks show significant differences between what and what??:  would noty I be possible to have anova and letters?

Our response:

We wrote: “Asterisks over the bar show significance between F25Cd and either of F2C or F25Cd groups”. In (E), asterisks show significant differences to F2C group. In (F), asterisk over F25C indicates significant (p<0.05) difference to F2C group.”

in 3B asterisks indicates difference between F25Cold and F2C? same for 3

but what about 3E and 3F??

Our response:

We wrote: “Asterisks over the bar show significance between F25Cd and either of F2C or F25Cd groups” – this implies to all figures with the bars. In addition, in (E), asterisks show significant differences to F2C group. In (F), asterisk over F25C indicates significant (p<0.05) difference to F2C group.”

In figure 4 the horizontal lines have not the same width… (in 4A and 4C).

 Our response:

The width is arbitrary, just to indicate connecting comparison groups.

Figures 4, 5, 6: what about F2? is that F2C??

Our response:

Corrected.

Figure 5;  here again the asterisk is supposed to indicate significant differences between the indicated group and two other groups. By default, the other groups are not significantly different? It would be better to have letters or clear horizontal line indicating the pairwise comparison.

Our response:

That is correct, all others are not significantly different from each other.

in figure 6 you indicate “Asterisks over the bar show significance (p<0.05) between the indicated group and two other groups.”  but the horizontal line would suggest a comparison between F25Cold and F2C… it is confusing.

same remark for fig7

Why is figure 7 divided in two rectangles??

Our response:

We corrected to: Asterisks over the bar show significance (t-test, two-sample assuming unequal variances; p<0.05) between F25Cd and any of two other groups. We removed the boxes.

Figure 10. write F25Cd vs. F2 and not F25Cd _F2. same for F25Cold vs. F25C. I note that you write F2_ct and not F2C for methylomes…..Here again another nomenclature….

 Our response:

Corrected.

line 391 you write “In contrast, in the F25Cd vs. F25C group comparison, F25Cd samples were

not as clearly separated from F25C group.” I do not think this is true of Fig11D and 11F.

 Our response:

It is actually the case for CG and CHG, so we have added it: “In contrast, in the F25Cd vs. F25C group comparison, F25Cd samples were not as clearly separated from F25C group, especially in CG and CHG context.”

line 389-391: indicate figure panels readers are supposed to look at.

 Our response:

Done

Figure 12/ rewrite the titles of each anel. Do not use “_” for versus.

why for F25_CD you have F25_Cd6 but not F25_Cd3?

in some panels you have numbered 1 to 5 and in other you have numbered your samples 1 to 10 (even though they do not correspond to the same lines)!! keep 1 to 5.

Our response:

Done

I have small remarks to make:

line 88-89: is that correct? I think you did also the analysis at generations 10, 15 and 20? Am I right?

Our response:

Not relevant any more as we deleted phenotypic data.

You wrote “Since they can not escape stress, they have to develop mechanisms of remembering stress exposures somatically and passing it to the progeny.”

write « . They have to developed mechanisms of remembering stress exposures somatically and passing it to the progeny.”

Our response:

Done.

I do not agree with the fact that plants “had to”…

Our response:

Corrected.

line 11 Arabidopsis thaliana in italics

Our response:

Done

line 18: maybe you need to explain what are CHG and CHH contexts for readers not at ease with genetics. What does H stand for?

Our response:

Done. We added information in the Intro: DNA methylation occurs in CG, CHG and CHH contexts in plants where C is a cyto-sine, G is a guanine and H represents the nucleotides A, T or C.

Reviewer 3 Report

Comments and Suggestions for Authors

Personally, I think that the manuscript is definitely too long and technically poorly made. The charts are too small and illegible. Most of them require changes. Therefore, it does not arouse the reader's interest. The results obtained are not particularly interesting or revealing. What new things do they bring to science? What could be the practical aspect of their use?

The graphs in Figure 1 are very difficult to read. There are no error bars or statistical differences visible. I suggest enlarging them and placing them one below the other.

I suggest that in Figure 6 A-C the scale be made from 0. When presenting the results in this way, the errors seem to be very large.

In Figure 7 E-H, please replace the gray font with black.

The dendrograms in Figures 10 and 11 are completely unreadable. You can't see the markings that are marked in red and blue.

Figure 13 shows no signatures at all. The letters are too small. The same situation is on the Figure 14.

The discussion is very long, I would suggest shortening it.

Author Response

Reviewer #3

Personally, I think that the manuscript is definitely too long and technically poorly made. The charts are too small and illegible. Most of them require changes. Therefore, it does not arouse the reader's interest. The results obtained are not particularly interesting or revealing. What new things do they bring to science? What could be the practical aspect of their use?

Our response:

Most of the figures are redone.

The graphs in Figure 1 are very difficult to read. There are no error bars or statistical differences visible. I suggest enlarging them and placing them one below the other.

Our response:

This is now deleted.

I suggest that in Figure 6 A-C the scale be made from 0. When presenting the results in this way, the errors seem to be very large.

Our response:

Done

In Figure 7 E-H, please replace the gray font with black.

Our response:

Done

The dendrograms in Figures 10 and 11 are completely unreadable. You can't see the markings that are marked in red and blue.

Our response:

This is now redone.

Figure 13 shows no signatures at all. The letters are too small. The same situation is on the Figure 14.

Our response:

This is now redone.

The discussion is very long, I would suggest shortening it.

Our response:

Shortened.

Round 2

Reviewer 1 Report

Comments and Suggestions for Authors

Since my first review mainly focused on morphological phenotypes features, since it is deleted in this revision which I understand due to the data provided, I regret honestly added phenotype data in this second version, rarely provided in molecular papers and making the originality of the idea of the first version.

However, some points of the present version have to be clarified:

-       - For my point 2, the sentences of the introduction “The environmentally induced and inherited epigenetic marks can facilitate plants’ adaptation to the changing environments. It can for example cause short-term microevolution in clonal plants [22]” have to be rephrased as, bespite of the article cited, “adaptation” related with “epigenetic” is nowadays very discussed, “short-term” related with “microevolution” too.

-      -  For my major point 4 (your 4.4), include your interesting response, even in shorter sentences, somewhere in your text.

-       - For my minor point 1, still italics are missing in the present version; moreover Arabidopsis has several subtaxa also used in molecular biology, in order to avoid misunderstandings please use precise taxa names overall your text.

-       - For my minor point 11, your (new) figures 9-10 still contain too many abbreviations making this part very difficult to understand rapidly especially for non-specialists, which is the target of figures.

-       - For the same reasons, in the abstract, CHG and CHH should be in full letters,

-       - I cannot see the corrections of my minor point “18 At the end of 3.3.2, precise the taxa involved in the references 63 and 64”.

-       - For the 25 generations (about 3 months for each as you wrote), put somewhere “for a total of xxx years of experiments” or something similar.

-         

Author Response

However, some points of the present version have to be clarified:

-       - For my point 2, the sentences of the introduction “The environmentally induced and inherited epigenetic marks can facilitate plants’ adaptation to the changing environments. It can for example cause short-term microevolution in clonal plants [22]” have to be rephrased as, bespite of the article cited, “adaptation” related with “epigenetic” is nowadays very discussed, “short-term” related with “microevolution” too.

Our response

Our apologies. While we agree with the latter part, we should have written “It can for example lead to microevolution in clonal plants in a relative short time”, it is not entire clear to us what is wrong with the first part “The environmentally induced and inherited epigenetic marks can facilitate plants’ adaptation to the changing environments.” Adaptation to changing environment may require changes in the physiology that are triggered in part by epigenetic regulations. It is plausible to make a connection between inheritance of epigenetic marks and adaptation.

-      -  For my major point 4 (your 4.4), include your interesting response, even in shorter sentences, somewhere in your text.

Our response

This is now included in the Discussion.

-       - For my minor point 1, still italics are missing in the present version; moreover Arabidopsis has several subtaxa also used in molecular biology, in order to avoid misunderstandings please use precise taxa names overall your text.

Our response

Our apologies, we have made it italic in several more cases we found (6 or 7). We have introduced “ecotype Columbia” into the Abstract. Frankly, I have been working with Arabidopsis since 1996, and I am not familiar with subtaxa. I use this specific line of Columbia cultivar since 1996.

-       - For my minor point 11, your (new) figures 9-10 still contain too many abbreviations making this part very difficult to understand rapidly especially for non-specialists, which is the target of figures.

Our response

We have added additional information to Figure captions.

-       - For the same reasons, in the abstract, CHG and CHH should be in full letters,

Our response

We have added this information.

-       - I cannot see the corrections of my minor point “18 At the end of 3.3.2, precise the taxa involved in the references 63 and 64”.

Our response

We actually did add it; “A. thaliana” was inserted in the beginning of the sentence (these are references 56 and 57 now). We have attached the version with tracked changes to show it last time we submitted it.

-       For the 25 generations (about 3 months for each as you wrote), put somewhere “for a total of xxx years of experiments” or something similar.

Our response

Yes, we added the information that it took us about 9 years (it was not always possible to grow it back-to-back).